# Improved Approximations for Hard Graph Problems using Predictions

**Anders Aamand** [* 1]   **Justin Y. Chen** [2]   **Siddharth Gollapudi** [3]   **Sandeep Silwal** [4]   **Hao WU** [5]

## Abstract

We design improved approximation algorithms for NP-hard graph problems by incorporating predictions (e.g., learned from past data). Our prediction model builds upon and extends the $\varepsilon$-prediction framework by Cohen-Addad, d'Orsi, Gupta, Lee, and Panigrahi (NeurIPS 2024). We consider an edge-based version of this model, where each edge provides two bits of information, corresponding to predictions about whether each of its endpoints belong to an optimal solution. Even with weak predictions where each bit is only $\varepsilon$-correlated with the true solution, this information allows us to break approximation barriers in the standard setting. We develop algorithms with improved approximation ratios for MaxCut, Vertex Cover, Set Cover, and Maximum Independent Set problems (among others). Across these problems, our algorithms share a unifying theme, where we separately satisfy constraints related to high degree vertices (using predictions) and low-degree vertices (without using predictions) and carefully combine the answers.

## 1. Introduction

Learning-augmented algorithms have recently emerged as a popular paradigm for beyond worst-case analysis via incorporating machine learning into classical algorithm design. By utilizing predictions, e.g. those learned from past or similar instances, prior works have improved upon worst-case bounds for competitive ratios for online algorithms (Lykouris & Vassilvitskii, 2021; Mitzenmacher & Vassilvitskii, 2022), space usage in streaming applications (Hsu et al., 2019; Jiang et al., 2020; Chen et al., 2022b), and

*Authors listed alphabetically. [1]University of Copenhagen [2]MIT [3]UC Berkeley [4]UW-Madison [5]University of Waterloo. Correspondence to: Anders Aamand <aa@di.ku.dk>, Justin Y. Chen <justc@mit.edu>, Siddharth Gollapudi <sgollapu@berkeley.edu>, Sandeep Silwal <silwal@cs.wisc.edu>, Hao WU <hao.wu1@uwaterloo.ca>.

*Proceedings of the $42^{nd}$ International Conference on Machine Learning*, Vancouver, Canada. PMLR 267, 2025. Copyright 2025 by the author(s).

running times for classical algorithms (Mitzenmacher & Vassilvitskii, 2022; Chen et al., 2022a), among many other highlights.

We focus on a recent line of work on improving approximation guarantees for fundamental NP-hard problems. The goal of this line of work is to use predictions to circumvent hardness of approximation results in the worst case (under standard complexity conjectures). One compelling motivation for learning-augmented algorithms for NP-hard problems is the following: if we run a heavy-duty computation on a single input, such as running a commercial integer linear programming solver, can we avoid repeated computation for a *similar* future inputs? Can we augment classical approximation algorithms with insights learned from an expensive computation to warm-start any future calls to the algorithm? More generally, how can we incorporate noisy information, which is weakly correlated with an optimal solution, into the algorithm design pipeline for NP-hard problems?

To formalize this, the starting point of our paper is the recent work of Cohen-Addad et al., which introduced the $\varepsilon$-accurate vertex prediction model for the MaxCut problem. Their model provides the algorithm designer with vertex predictions: one bit for the side of the MaxCut each vertex lies in. Every bit is equal to the correct answer with probability $1/2 + \varepsilon$.

We are motivated by three follow up directions. The first direction explores the setting where we are optimizing a global objective function (as in MaxCut) with additional *edge-constraints*. For example, this captures the classic vertex-cover or independent set problems.

> *Can we generalize the setting of Cohen-Addad et al. to obtain learning-augmented algorithms for NP-hard graph optimization problems with edge constraints?*

This naturally leads us to introduce *edge predictions* as an extension of the vertex prediction model (Cohen-Addad et al., 2024). In our edge prediction model, every edge is equipped with i.i.d. bits for each variable participating in the edge constraint. Similar to the vertex prediction model, our predicted bits are also $\varepsilon$-correlated with the ground truth

| Problem | Classical Approx. | Our Learning-Augmented Approx. with $\varepsilon$-correlated Predictions | Reference |
|---|---|---|---|
| Vertex Cover | 2 | $2 - \Omega\left(\frac{\log\log 1/\varepsilon}{\log 1/\varepsilon}\right)$ | Theorem 4.1. |
| Weighted Vertex Cover | 2 | $2 - \Omega\left(\frac{\log\log 1/\varepsilon}{\log 1/\varepsilon}\right)$ | Theorem A.1. |
| Set Cover | $O(\log n)$ | $O(\log 1/\varepsilon)$ | Theorem B.2. |
| Maximum Independent Set | $\Omega\left(\frac{\log n}{n \log\log n}\right)$ | $\Omega\left(\frac{\varepsilon^2}{\log\log 1/\varepsilon}\right)$ | Theorem 5.4. |
| MaxCut | $\alpha_{GW} \approx .8786$ | $\alpha_{GW} + \tilde{\Omega}(\varepsilon^2)$ | Theorem C.2. |

*Table 1.* Summary of our main results. All approximation factors are for multiplicative approximation. Note that the first three problems are *minimization* problems, so we want the approximation factor to be as as small as possible. The last two problems are *maximization* problems, so we desire a large approximation factor. The bounds listed above are consistent with this distinction. $\alpha_{GW}$ refers to the Goemans-Williamson constant (Goemans & Williamson, 1995). For all of these problems, the stated classical approximation bounds are tight assuming $P \neq NP$ and the Unique-Games conjecture (for MaxCut).

(see Definition 1.1 for a formal definition of our model), meaning each bit is equal to the correct assignment (e.g. is the vertex adjacent to the edge in the optimal vertex cover or not) with probability $1/2 + \varepsilon$. The second direction explores the power of our newly introduced edge predictions.

> *What is the power of edge predictions in the learning-augmented setting for NP-Hard graph problems? Can we understand a general framework for designing augmented graph algorithms with edge-predictions?*

It is important to note that several concurrent or follow-up works to (Cohen-Addad et al., 2024) have appeared, such as (Bampis et al., 2024; Ghoshal et al., 2025; Antoniadis et al., 2024; Braverman et al., 2024) which also study graph problems with edge constraints, albeit with only vertex predictions, as opposed to edge predictions introduced in our work (see Section 3 for a detailed comparison). Thus, we also ask:

> *Can our edge predictions lead to improved approximation algorithms beyond vertex predictions for fundamental NP-Hard graph problems?*

**Our Contributions**  We individually address the three directions above and our contributions towards them.

**Direction 1:** For the first direction, we introduce *edge predictions* for NP-hard graph problems. As a prototypical example, the following is our prediction model for the Vertex Cover problem. Recall in the Vertex Cover problem we want to find the smallest subset of vertices such that every edge is adjacent to some vertex in the subset.

**Definition 1.1** (VC Prediction Model). Given an input unweighted and undirected graph $G$, we fix some optimal vertex cover $\mathcal{C}$ of $G$. Every edge $e = (u, v)$ outputs two bits $(b_u(e), b_v(e))$ of predictions. If $u \in \mathcal{C}$ then $b_u(e) = 1$ with probability $1/2 + \varepsilon$ and 0 with probability $1/2 - \varepsilon$. Similarly if $u \notin \mathcal{C}$ then $b_u(e) = 0$ with probability $1/2 + \varepsilon$ and 1 with probability $1/2 - \varepsilon$. $b_v(e)$ is similarly sampled and all predictions are i.i.d. across all edges and bits.

We appropriately generalize the above edge prediction model to a wide range of problems, including weighted Vertex Cover, Set Cover, Maximum Independent Set, and MaxCut. Our prediction models are unified under a similar theme (the natural analogues of Definition 1.1 to other NP-hard problems): every edge (or a hyper-edge in the context of Set Cover) gives a single bit of information for each vertex that it is adjacent to. These bits are $\varepsilon$-correlated with a true underlying assignment. For brevity, we omit their definitions here and refer to Section 2.

**Direction 2:** We demonstrate the power of our edge predictions by giving improved approximation bounds, which go beyond known approximation barriers, for many fundamental NP-hard graph optimization problems. Our results are summarized in Table 1.

For Vertex Cover and weighted Vertex Cover, we obtain an algorithm with approximation ratio $2 - f(\varepsilon)$ for $f(\varepsilon) = \Omega\left(\frac{\log\log 1/\varepsilon}{\log 1/\varepsilon}\right)$. Note that $f$ is a much faster growing function than any polynomial in $\varepsilon$ in the sense that $f(\varepsilon) = \Omega(\varepsilon^C)$ for any constant $C$. For any constant $\varepsilon$, this implies a strictly smaller constant than 2. Such an algorithm in the standard setting without predictions, would imply that the Unique Games Conjecture is false (Khot & Regev, 2003).

For Set Cover we obtain a constant approximation factor for any constant $\varepsilon > 0$, in comparison to the classical setting where a $O(\log n)$ factor is tight (again assume $P \neq NP$) (Dinur & Steurer, 2014). We defer the discussion of Maximum Independent Set and MaxCut.

We highlight that in all of our main results, the notion of heavy vertices (those with large degree for an appropriately defined threshold) vs light vertices (those with degree smaller than the threshold) is a central theme. In fact, our methodology in improving approximation ratios for different NP-hard problems can appropriately be captured by the high-level algorithm design framework of Figure 1. In this framework, high degree vertices use the majority of their prediction bits $b_v(e)$ for each edge $e$ incident to $v$ of incident edges to decide if they should be included in the solution or not.

Then, all high degree vertices as well as low-degree vertices having an edge to a high-degree vertex are removed from the graph, and we run an appropriate solver on the remaining low-degree graph. The idea is that on the one hand, for the high degree vertices, the majority vote is a very accurate prediction, and on the other hand, for graphs with bounded maximum degree, there often exist polynomial time algorithms with better approximation guarantees.

We emphasize that the framework is as a guiding principle, but the nuances of each problem must be separately dealt with individually. Nevertheless, we find it useful to highlight this connecting thread among our different algorithms, as it demonstrates the wide applicability of our prediction model.

Some of the non-trivial parts that Figure 1 glosses over which must be handled on a problem-by-problem basis are:

- False-positive vertices: (high degree vertices $v$ in Line 2 on which the majority incorrectly sets $b'_v = 1$). These are problematic since there are potentially $\Omega(n)$ many false-positive vertices, which may artificially increase the size of the solution we output. For example in the Vertex Cover problem, if OPT is sublinear in $n$, we would not get any bounded competitive ratio, let alone a ratio that is smaller than 2.

- False-negative vertices (high degree vertices $v$ in Line 2 on which the majority incorrectly sets $b'_v = 0$). On the flip side, if we fail to identify a vertex that is actually in OPT, we may not satisfy all the edge constraints, even among only those that high-degree vertices participate in.

- Boundary effects: We must be careful when combining an assignment on high-degree vertices (obtained via voting in line 2 of the framework in Figure 1) and an assignment for low-degree vertices (computed using a solver on the low-degree vertices in line 4 of Figure 1). These two solutions may conflict with each other. For example in

---

A Framework for Incorporating Edge-Predictions in Learning-Augmented Algorithms.

**Input:** Graph $G = (V, E)$, edge predictions $(b_u(e), b_v(e)) \in \{0, 1\}^2$ for every edge $e = (u, v) \in E$.

**Output:** A bit $b'_v$ for all $v \in V$ satisfying certain edge constraints.

1. Set an appropriate vertex degree threshold $\Delta$.

2. Loop over vertices $v$ satisfying $\deg(v) \geq \Delta$ and set $b'_v = \text{Majority}(\{b_v(e)\}_{e \in E, v \in e})$.

3. Remove all high-degree vertices from $G$, all low degree vertices which share a constraint with any high-degree vertex, as well as all the constraints corresponding to their edges

4. Run an appropriate solver on the remaining low-degree graph.

*Figure 1.* A high-level description of the underlying structure common among our algorithms.

---

the Independent Set problem, an edge may be adjacent to both low-degree and high-degree vertices, and both steps (lines 2 and 4) may select both endpoints which would violate the constraints of the problem.

We refer to the individual problem sections for more details.

To complement our theoretical results, we also experimentally test our algorithm compared to baselines which only run a worst-case approximation algorithm or only follow the predictions for the problem of Maximum Independent Set.

**Direction 3:** Lastly for the third direction, we show separations between our edge predictions and previously considered vertex predictions for Vertex Cover (Antoniadis et al., 2024), Maximum Independent Set (Braverman et al., 2024), and MaxCut (Cohen-Addad et al., 2024).

We begin by discussing the Vertex Cover problem. Recent work by Antoniadis, Eliáš, Polak, and Venzin (2024) explores the learning-augmented setting of Vertex Cover with vertex predictions. Their algorithm achieves an approximation ratio of $1 + (\eta^+ + \eta^-)/\text{OPT}$, where $\eta^+$ and $\eta^-$ are the number of false positive and false negative vertices, respectively in a given predicted solution. They further prove that, under the Unique Games Conjecture and the condition $(\eta^+ + \eta^-)/\text{OPT} \leq 1$, this bound is tight. However, this result necessarily requires strong assumptions about the prediction oracle to go beyond the known approximation results

without prediction. For instance, if we apply this result to the $\varepsilon$-correlated vertex prediction model of (Cohen-Addad et al., 2024), where each vertex is predicted to belong to an optimal solution with probability $1/2 + \varepsilon$ independently, the ratio $1 + (\eta^+ + \eta^-)/\text{OPT}$ can become arbitrarily larger than 2, which is significantly larger than the classic factor of 2-approximation.

Thus, a meaningful instantiation of their theorem requires an implicit assumption that *both* the number of false positive and false negative vertices are bounded. We remark that in our edge prediction model, we make no such assumptions about the number of false positive or negative vertices. Rather, we show that these terms can be related to the optimal vertex cover size, through a technical charging argument, detailed in Section 4.

A qualitatively similar barrier for vertex predictions arises in the Maximum Independent Set (MIS) problem. Braverman, Dharangutte, Shah, and Wang (2024) similarly address MIS with the same vertex prediction model of (Cohen-Addad et al., 2024): for each vertex, the oracle predicts its membership in a fixed optimal MIS solution with probability $1/2 + \varepsilon$. Their algorithm achieves an approximation ratio of $\tilde{O}(\sqrt{\Delta}/\varepsilon)$, where $\Delta$ is the maximum degree of the graph. Note that in the worst case, this is an approximation ration of $\Omega(\sqrt{n})$ if there are high-degree nodes in the graph. In contrast, using edge predictions allow us to obtain a *constant* factor approximation for any fixed $\varepsilon$.

Lastly, we discuss MaxCut, the original problem studied in (Cohen-Addad et al., 2024). Here, Cohen-Addad et al. (2024) obtain a $\alpha_{GW} + \tilde{\Omega}(\varepsilon^4)$ approximation factor, whereas we obtain a larger advantage of $\alpha_{GW} + \tilde{\Omega}(\varepsilon^2)$ over $\alpha_{GW}$, the Goemans-Williamson (since $\varepsilon \in (0, 1), \varepsilon^4 \ll \varepsilon^2$).

In summary, the vertex prediction model and our edge prediction models are to a degree incomparable: we obtain information from every edge (i.e. we receive more bits as predictions than the vertex prediction model), but our approximation ratios are also stronger for MaxCut and MIS. For other problems such as Vertex Cover, we obtain a strictly smaller competitive ratio than 2, which the aforementioned results cannot without assuming bounds on the false positive and negative ratio among the predictions.

**Organization** The paper is organized as follows. Section 2 provides formal definitions of the studied problems. Section 3 reviews related work. Section 4 presents our algorithm for the Vertex Cover problem. Section 5 introduces our algorithm for the Maximum Independent Set problem. Section 6 shows experimental results for our algorithm in the context of Maximum Independent Set. Due to space constraints, our algorithms for the Weighted Vertex Cover, Set Cover, and Max Cut problems are included in Appendix A, Appendix B, and Appendix C, respectively. Our empirical results are in Section 6.

## 2. Preliminaries

We recall the definitions of the class NP-Hard problems studied here. Let $G = (V, E)$ be an undirected graph with $n = |V|$ vertices and $m = |E|$ edges.

- **Vertex Cover** Given an input graph $G$, the Vertex Cover (VC) problem seeks a minimum cardinality vertex subset such that every edge is incident to at least one vertex in the subset.

- **Weighted Vertex Cover** The weighted Vertex Cover (WVC) problem is an extension of the VC problem, where each vertex $v \in V$ is associated with a weight $w(v) \in \mathbb{R}^+$. The WVC problem seeks a vertex subset with minimum total weight such that every edge is incident to at least one vertex in the subset.

- **Set Cover** Let $\mathcal{U}$ be a finite set of elements, and let $S_1, \ldots, S_n \subseteq \mathcal{U}$ be a collection of subsets. The Set Cover (SC) problem seeks a collection of the minimum number of subsets whose union is $\mathcal{U}$. For reasons that will become clear when we clarify the prediction model for the SC problem, we view each subset $S_i$ as a "vertex" and each element $u \in \mathcal{U}$ as a hyperedge that spans the subsets containing $u$. Hence, without loss of generality, we assume $\mathcal{U} = [m]$.

- **Maximum Independent Set** Given an input graph $G$, the Maximum Independent Set (MIS) problem seeks a maximum cardinality vertex subset such that no two vertices in the subset are adjacent.

- **MaxCut** Given an input graph $G$, a max cut is a partition $(S, T)$ of the vertices $V$ such that the number of edges between $S$ and $T$ is maximized.

## 3. Related Work

**(Weighted) Vertex Cover.** Khot and Regev (2008) proved that, assuming the Unique Games Conjecture and P≠NP, the (Weighted) Vertex Cover problem cannot be approximated within a factor of $2 - \varepsilon$ for any $\varepsilon > 0$. Antoniadis, Eliáš, Polak, and Venzin (2024) studied approximation algorithms for the weighted version of this problem under a prediction oracle that provides a predicted vertex set $\hat{X}$ of the optimal solution $X^*$. This framework generalizes the vertex-based model, which predicts the membership of each vertex independently. Their algorithm achieves an approximation ratio of $1 + (\eta^+ + \eta^-)/\text{OPT}$, where $\eta^+ \doteq w(\hat{X} \setminus X^*)$ is the total weight of false positives, and $\eta^- \doteq w(X^* \setminus \hat{X})$ is the total weight of false negatives. They further show that, assuming the Unique Games Conjecture and $(\eta^+ + \eta^-)/\text{OPT} \leq 1$, this bound is tight.

**Set Cover** Dinur and Steurer (2014) proved that approximating the Set Cover problem within a factor of $(1-\varepsilon)\ln m$ is NP-hard for any $\varepsilon > 0$. For instances where each set has size at most $\Delta$, there exist $(1 + \ln\Delta)$-approximation algorithms (Johnson, 1974; Lovász, 1975; Chvátal, 1979). On the other hand, Trevisan (2001) showed that, for such instances, the problem is hard to approximate within a factor of $\ln\Delta - O(\ln\ln\Delta)$ unless P=NP.

**Maximum Independent Set (MIS).** The MIS problem is NP-hard to approximate within a factor of $n^{1-\delta}$ for any $\delta > 0$ (Håstad, 1996b). Braverman et al. (2024) attempt to bypass this barrier using the vertex prediction model. Their algorithm achieves an approximation ratio of $\tilde{O}(\sqrt{\Delta}/\varepsilon)$, where $\Delta$ is the maximum degree of the graph.

**Max Cut.** For the classical MAXCUT problem, the celebrated work of Goemans and Williamson (1995) achieved an approximation factor of $\alpha_{GW} \approx 0.878$, which is the best possible under the Unique Games Conjecture (UGC).

Recent research has explored methods to surpass this computational barrier using machine learning oracles. Cohen-Addad, d'Orsi, Gupta, Lee, and Panigrahi (2024) studied MAXCUT under a vertex-based prediction model, where the oracle predicts each vertex's label for the MAXCUT solution with probability $1/2 + \varepsilon$. Their algorithm achieves an approximation ratio of $\alpha_{GW} + \tilde{\Omega}(\varepsilon^4)$. In contrast, our edge-based prediction model improves this approximation ratio to $\alpha_{GW} + \tilde{\Omega}(\varepsilon^2)$.

Ghoshal, Markarychev, and Markarychev (2025) also investigated the MAXCUT problem under the vertex-based prediction model. Their algorithms achieve an approximation ratio of $1 - O\left(\frac{1}{\varepsilon\sqrt{2m/n}}\right)$ for unweighted graphs and an error bound of at most $\text{OPT} - \frac{1}{\varepsilon}\sqrt{n\sum_{i,j} w_{i,j}^2}$ for the weighted case.

Bampis, Escoffier, and Xefteris (2024) demonstrated that the $\alpha_{GW}$ barrier can be surpassed for a restricted class of dense graphs under the vertex-based prediction model. Instead of querying the prediction oracle for every vertex, their algorithm queries only a sampled set $S$ of size $\Theta\left(\frac{\ln n}{\alpha^3 \delta^4}\right)$. For a graph with edge density $\delta = \frac{2|E|}{n(n-1)}$, their polynomial-time algorithm achieves an approximation factor of $1 - \alpha - 8\frac{error}{\delta|S|}$, where *error* denotes the number of mispredicted vertices in $S$. Although their approach requires only a sublinear number of predictions, their approach necessitates the error rate to be close to 0 to surpass the $\alpha_{GW}$ barrier.

**Predictions for Other NP-Hard Problems** Lastly, we remark that (Ergun et al.) and (Gamlath et al., 2022) have also studied learning-augmented algorithms for NP-hard

clustering problems where noisy cluster labels are revealed.

## 4. Vertex Cover

In this section, we present Algorithm 1 for VC under the prediction model of Definition 1.1, achieving an approximation ratio better than 2.

**Theorem 4.1.** *Algorithm 1 returns a vertex cover with an approximation ratio of $2 - \Omega\left(\frac{\log\log 1/\varepsilon}{\log 1/\varepsilon}\right)$ in expectation.*

*Remark.* For any $C > 0$, the bound $2 - \varepsilon^C \in 2 - \Omega\left(\frac{\log\log 1/\varepsilon}{\log 1/\varepsilon}\right)$ holds for sufficiently small $\varepsilon$, implying our approximation factor surpasses $2 - \text{poly}(\varepsilon)$. Moreover, assuming the Unique Games Conjecture and P$\neq$NP, VC remains inapproximable within $2 - \varepsilon$ for any $\varepsilon > 0$ (Khot & Regev, 2008).

---

**Algorithm 1** LearnedVC

---

1: $S_0 \leftarrow \varnothing, \Delta \leftarrow 100\log(1/\varepsilon)/\varepsilon^2$
2: **for** $v \in V$ with $\deg(v) \geq \Delta$ **do**
3:    $m_v \leftarrow \text{Majority}(\{b_v(e)\}_{e \in E, v \in e})$       $\triangleright\, 0$ if tie
4: **end for**
5: **for** $v$ with $\deg(v) \geq \Delta$ **do**
6:    **if** $m_v = 1$ and all neighbors $u$ of $v$ satisfy $\deg(u) \geq \Delta$ and $m_u = 1$ **then**
7:       Skip $v$
8:    **else if** $m_v = 1$ **then**
9:       $S_0 \leftarrow S_0 \cup \{v\}$
10:   **else if** $m_v = 0$ **then**
11:      $S_0 \leftarrow S_0 \cup \{N(v)\}$   $\triangleright\, N(v)$ is the neighboring vertices of $v$
12:   **end if**
13: **end for**
14: $S_1 \leftarrow$ 2-approximate VC for heavy-heavy and heavy-light edges not covered by $S_0$
15: $S_2 \leftarrow$ a $\left(2 - 2\frac{\log\log\Delta}{\log\Delta}\right)$-approximate VC for edges not covered by $S_0 \cup S_1$
16: $S \leftarrow S_0 \cup S_1 \cup S_2$
17: **Return** $S$       $\triangleright$ Our VC approximation

---

**Algorithm** Given a threshold $\Delta \in \Theta(1/\varepsilon^2\ln(1/\varepsilon))$, Algorithm 1 classifies edges into three types:

**Definition 4.2.** For an edge $e = (u, v) \in E$ with $\deg(u) \leq \deg(v)$, we define:

$$\begin{cases} \text{heavy-heavy} & \text{if } \deg(u) \geq \Delta, \\ \text{heavy-light} & \text{if } \deg(v) \geq \Delta \text{ and } \deg(u) < \Delta, \\ \text{light-light} & \text{if } \deg(v) < \Delta. \end{cases}$$

Algorithm 1 proceeds in three stages. In the first stage, it tries to cover all heavy-heavy and heavy-light edges using

a vertex subset $S_0$ based on predicted information. Specifically, for each vertex $v$ with $\deg(v) \geq \Delta$, a predicted bit $m_v$ is computed. If $m_v = 1$, indicating $v$ is likely in the optimal solution ($\mathcal{C}$), then $v$ is added to $S_0$. Otherwise, all its neighbors $N(v)$ are added to $S_0$, since if $v \notin \mathcal{C}$, it must be that $N(v) \subseteq \mathcal{C}$.

An exception in stage one occurs when a vertex $v$ and all its neighbors $u \in N(v)$ have predicted bits $m_v = 1$ and $m_u = 1, \forall u \in N(v)$. Since $v \in \mathcal{C}$ and $N(v) \subseteq \mathcal{C}$ cannot hold simultaneously, Algorithm 1 defers the decision on $v$. Instead, in stage two (Line 14), it applies a conventional 2-approximation algorithm (Williamson & Shmoys, 2011) to identify a set $S_1$ such that $S_0 \cup S_1$ covers all heavy-heavy and heavy-light edges.

After the first two stages, the remaining uncovered edges are light-light edges. Algorithm 1 then applies a $\left(2 - 2\frac{\log \log \Delta}{\log \Delta}\right)$-approximate VC algorithm (Halperin, 2002) to cover them.

**Analysis**    To prove Theorem 4.1, we establish the following lemmas, each bounding the size of 'bad' events throughout the execution of Algorithm 1.

**Lemma 4.3** (False Vertices due to Prediction)**.**

$$\mathbb{E}\left[|S_0 \setminus \mathcal{C}|\right] \leq \varepsilon^{10} \cdot |\mathcal{C}|.$$

**Lemma 4.4** (Fix for Predicted Set)**.** *Let* $\mathcal{C}_{\geq \Delta} \doteq \{v \in \mathcal{C} : \deg(v) \geq \Delta\}$. *Then*

$$\mathbb{E}[|S_1|] \leq 2 \cdot \varepsilon^{200} \cdot |\mathcal{C}_{\geq \Delta}|.$$

**Lemma 4.5** (Cover for Light Edges)**.** *Let* $\mathcal{C}_{<\Delta} \doteq \mathcal{C} \setminus \mathcal{C}_{\geq \Delta}$. *Then*

$$\mathbb{E}[|S_2|] \leq \mathbb{E}\left[|\mathcal{C}_{<\Delta} \setminus (S_0 \cup S_1)| \cdot \left(2 - 2\frac{\log \log \Delta}{\log \Delta}\right)\right].$$

The complete proofs of them are included in Appendix E. Here, we provide brief, intuitive explanations.

**Lemma 4.3** states that Algorithm 1 does not add too many vertices outside $\mathcal{C}$ into $S_0$. This can occur in two cases:

Case One: A vertex $v \in \mathcal{C}_{\geq \Delta}$ with $m_v = 0$ causes its neighbors $N(v)$, including vertices outside $\mathcal{C}$, to be added to $S_0$. To bound the expected number of such vertices, it suffices to show $\deg(v) \cdot \Pr[m_v = 0] \in o(1)$. Using standard concentration inequalities, we establish that $\Pr[m_v = 0] \in o(1/\deg(v))$. Summing over all $v \in \mathcal{C}_{\geq \Delta}$ then bounds the number of added vertices by $o(|\mathcal{C}_{\geq \Delta}|)$.

Case Two: A vertex $v \notin \mathcal{C}$ with $\deg(v) \geq \Delta$ and $m_v = 1$ is directly added to $S_0$. To count such vertices, we use a "charging argument": we charge the cost of adding $v$ to a neighboring vertex $u \notin S_0$. Since $v \notin \mathcal{C}$, it must be that

$u \in \mathcal{C}$. Through careful analysis, we show that each $u \in \mathcal{C}$ is charged at most $\deg(u)$ times, with each charge occurring with probability $o(1/\deg(u))$. Summing over all $u \in \mathcal{C}$ bounds the total "charging cost" by $o(|\mathcal{C}|)$.

**Lemma 4.4** bounds the size of $S_1$. First, we show that $\mathcal{C}_{\geq \Delta} \cup S_0$ covers all heavy-heavy and heavy-light edges. Thus, adding $\mathcal{C}_{\geq \Delta} \setminus S_0$ to $S_0$ ensures full coverage of these edges.

Next, we prove that $\mathbb{E}\left[|\mathcal{C}_{\geq \Delta} \setminus S_0|\right] \leq \varepsilon^{200} \cdot |\mathcal{C}_{\geq \Delta}|$. This follows because if $v \in \mathcal{C}_{\geq \Delta}$ is not in $S_0$, then either $m_v = 0$, which happens with probability $\Pr[m_u = 0] = O(\varepsilon^{200})$, or it has been skipped at step 7. Since $v$ has a neighbor $u \notin \mathcal{C}$ (otherwise, $v$ could be removed from $\mathcal{C}$), and skipping occurs only if $\deg(u) \geq \Delta$ and $m_u = 1$, the probability of skipping $v$ is bounded by $\Pr[m_u = 1] = O(\varepsilon^{200})$.

Finally, the size bound of $S_1$ in Lemma 4.4 follows from $S_1$ being a 2-approximate solution.

**Lemma 4.5** follows from: 1) $\mathcal{C}_{<\Delta}$ covers all light-light edges, and 2) the edges remaining after selecting $S_0 \cup S_1$ are exclusively light-light edges. Thus, $\mathcal{C}_{<\Delta} \setminus (S_0 \cup S_1)$ forms a valid cover for these edges. Consequently, the expected size of $S_2$ is at most $\left(2 - 2\frac{\log \log \Delta}{\log \Delta}\right)$ times the size of $\mathcal{C}_{<\Delta} \setminus (S_0 \cup S_1)$.

*Remark* 4.6. We assume full independence in the prediction model of Definition 1.1 to simplify the analysis. In fact, 4-wise independence suffices to the lemmas. See the discussion after the formal proof of Lemma 4.3 in Appendix E for example.

*Proof Sketch of Theorem 4.1.* Since $S_2$ covers all edges not covered by $S_0 \cup S_1$, $S \doteq S_0 \cup S_1 \cup S_2$ is a VC. The expected cost is bounded by

$$\mathbb{E}\left[|S|\right] \leq \mathbb{E}\left[|S_0|\right] + \mathbb{E}\left[|S_1|\right] + \mathbb{E}\left[|S_2|\right].$$

$\mathbb{E}\left[|S_0|\right]$ can be decomposed into $\mathbb{E}\left[|S_0 \cap \mathcal{C}|\right] + \mathbb{E}\left[|S_0 \setminus \mathcal{C}|\right]$. Bounding $\mathbb{E}\left[|S_1|\right]$ by $O(\varepsilon^{200} \cdot |\mathcal{C}_{\geq \Delta}|)$ with Lemma 4.4, $\mathbb{E}\left[|S_0 \setminus \mathcal{C}|\right]$ by $\varepsilon^{10} \cdot |\mathcal{C}|$ with Lemma 4.3, and $\mathbb{E}\left[|S_2|\right]$ by $\mathbb{E}\left[|\mathcal{C} \setminus S_0| \cdot \left(2 - 2\frac{\log \log \Delta}{\log \Delta}\right)\right]$ with Lemma 4.5, and summing up the upper bounds of $\mathbb{E}\left[|S_1|\right]$, $\mathbb{E}\left[|S_0 \setminus \mathcal{C}|\right]$ and $\mathbb{E}\left[|S_0 \cap \mathcal{C}|\right] + \mathbb{E}\left[|S_2|\right]$, proving the theorem. $\qquad \square$

## 5. Maximum Independent Set

We now present our prediction-based algorithm for approximating maximum independent set. For a graph $G = (V, E)$, we denote $\alpha(G)$ the size of a maximum independent set of $G$. Our goal is to find an independent set which is not much smaller than $\alpha(G)$. In general, $\alpha(G)$ cannot be approximated in polynomial time within multiplicative $n^{1-\gamma}$

**Algorithm 2** Learned Maximum Independent Set

1: $\Delta \leftarrow 3\log(1/\varepsilon)/\varepsilon^2$
2: $V_{\leq\Delta} \leftarrow \{v \in V | d_v \leq \Delta\}$
3: $V_{>\Delta} \leftarrow V \setminus V_{\leq\Delta}$
4: $\mathcal{C}_1 \leftarrow$ an $\Omega\left(\left(\frac{\log\Delta}{\Delta\log\log\Delta}\right)\right.$-approximate maximum independent set of $G[V_{\leq\Delta}]$
5: **for** $v \in V_{>\Delta}$ **do**
6: $\quad m_v \leftarrow \text{Majority}(\{b_v(e)\}_{e \in E, v \in e})$ $\qquad \triangleright 0$ if tie
7: **end for**
8: $S \leftarrow \{v \in V_{>\Delta} \mid m_v = 1\}$
9: **while** $G[S]$ has an edge $e = (u, v)$ **do**
10: $\quad S \leftarrow S \setminus \{u, v\}$
11: **end while**
12: $\mathcal{C}_2 \leftarrow S$
13: **if** $|\mathcal{C}_2| \geq |\mathcal{C}_1|$ **then**
14: $\quad$ **return** $\mathcal{C}_2$
15: **end if**
16: **return** $\mathcal{C}_1$

for any fixed $\gamma > 0$ unless $ZZP = NP$[1] (Håstad, 1996a). However, for bounded degree graphs, there are approximation algorithms where the approximation factor depends only on the maximum degree. We will use the following theorem (Halldórsson, 1998) for handling such bounded degree graphs.

**Theorem 5.1** ((Halldórsson, 1998)). *Maximum independent set can be approximated within $\Omega(\frac{\log\Delta}{\Delta\log\log\Delta})$ with high probability in polynomial time for graphs with maximum degree at most $\Delta$.*

We assume access to edge predictions as before.

**Definition 5.2** (Prediction Model). Given an input unweighted and undirected graph $G$, we fix some maximum independent set $\mathcal{C}^*$ of $G$. Every edge $e = (u, v)$ outputs two bits $(b_u(e), b_v(e))$ of predictions. If $u \in \mathcal{C}^*$ then $b_u(e) = 1$ with probability $1/2 + \varepsilon$ and 0 otherwise. $b_v(e)$ is similarly sampled and all predictions are i.i.d. across all edges and bits.

With access to $\varepsilon$-accurate predictions, our algorithm essentially achieves approximation $\tilde{\Omega}(\varepsilon^2)$ (note that without predictions, we cannot hope for any constant factor approximation). It is presented in Algorithm 2 and works as follows. We define $\Delta = 3\log(1/\varepsilon)/\varepsilon^2$, and further, $V_{\leq\Delta} = \{v \in V \mid d_v \leq \Delta\}$, $V_{>\Delta} = V \setminus V_{\leq\Delta}$, $G_1 = G[V_{\leq\Delta}]$ and $G_2 = G[V_{>\Delta}]$. For $G_1$ we use the algorithms from Theorem 5.1 finding an approximate maximum independent set $\mathcal{C}_1$. On the other hand, for the graph $G_2$, for each vertex $v \in V_{>\Delta}$, we include it in a preliminary

[1]$ZZP$ is the class of problems that can be solved in expected polynomial time by a Las Vegas algorithm. The conjecture $P \neq NP$ is almost as strong as $ZZP \neq NP$

set $S$ using the majority vote of all the incident edges. Note that two adjacent notes in $G_2$ may both be included in $S$ during this process. To form an independent set we update $S$ via the following *clean-up* process: As long as the induced graph $G[S]$ contains an edge, we pick any such edge and remove both endpoints from $S$. We denote the resulting set after this removal process by $\mathcal{C}_2$. The algorithm outputs the independent set $\mathcal{C}$ defined to be whichever of $\mathcal{C}_1$ or $\mathcal{C}_2$ that has the largest cardinality.

For the analysis of this algorithm, we require the classic Caro-Wei bound on $\alpha(G)$ to 'certify' that the actual optimal solution is sufficiently large. This is crucial for our charging argument of Theorem 5.4.

**Lemma 5.3** (Caro-Wei). *Let $G = (V, E)$ be a graph and denote by $d_v$ the degree of $v \in V$. Then $\alpha(G) \geq \sum_{v \in V} \frac{1}{1+d_v}$.*

Our main theorem on Algorithm 2 is as follows.

**Theorem 5.4.** *Suppose $\varepsilon \leq 1/4$. The expected size of the maximum independent set output by Algorithm 2, is $\Omega\left(\alpha(G) \cdot \frac{\varepsilon^2}{\log\log 1/\varepsilon}\right)$*

*Proof of Theorem 5.4.* Define $T = V_{>\Delta} \setminus \mathcal{C}^*$ and call a vertex $v \in T$ *bad* if $v$ voted to be in $S$. We claim that the expected number of bad vertices is at most $\varepsilon\alpha(G)$. To see this, define $X_v = [v \text{ is voted into } S]$ for $v \in V_{>\Delta}$ and $X = \sum_{v \in T} X_v$. By Hoeffding's inequality over the voting, $\Pr[X_v = 1] \leq \exp(-2\varepsilon^2 d_v)$, and using the fact that $e^x \geq 2x$ for all $x \in \mathbb{R}$, we obtain that

$$\Pr[X_v = 1] \leq \frac{1}{\exp(\varepsilon^2 d_v) 2\varepsilon^2 d_v}.$$

Finally, using that $v \in V_{>\Delta}$ and thus has degree at least $\Delta$, it follows that $\exp(\varepsilon^2 d_v) \geq \varepsilon^{-3}$ and thus,

$$\Pr[X_v = 1] \leq \frac{\varepsilon}{2d_v} \leq \frac{\varepsilon}{1 + d_v}.$$

By Lemma 5.3 and linearity of expectation, it follows that the expected number of bad vertices is at most $\varepsilon \cdot \alpha(G)$, as claimed. Additionally, note that for $v \in V_{>\Delta} \cap \mathcal{C}^*$, by Hoeffding's inequality $\Pr[X_v = 0] \leq \exp(-2\varepsilon^2 d_v) \leq \varepsilon^6$. Now during the clean-up phase, whenever we remove two nodes from $S$, one of them must have been bad. Indeed, two such nodes were included by majority vote into $S$ but only one of them could have been in $\mathcal{C}^*$ since they are connected by an edge. Thus

$$\mathbb{E}[|\mathcal{C}_2|] \geq (1 - \varepsilon^6)|\mathcal{C}^* \cap V_{>\Delta}| - 2\varepsilon \cdot \alpha(G). \quad (1)$$

Moreover, by Theorem 5.1

$$\begin{aligned} \mathbb{E}[|\mathcal{C}_1|] &= \Omega\left(\frac{\log\Delta}{\Delta\log\log\Delta}\alpha(G_1)\right) \\ &= \Omega\left(\frac{\log\Delta}{\Delta\log\log\Delta} \cdot |\mathcal{C}^* \cap V_{\leq\Delta}|\right). \quad (2) \end{aligned}$$

Now

$$\mathbb{E}[|\mathcal{C}|] = \mathbb{E}[\max(|\mathcal{C}_1|, |\mathcal{C}_2|)] \geq \max(\mathbb{E}\,[|\mathcal{C}_1|], \mathbb{E}[|\mathcal{C}_2|]).$$

If $|\mathcal{C}^* \cap V_{\leq \Delta}| \geq |\mathcal{C}^*|/2 = \alpha(G)/2$, then (2) gives that

$$\mathbb{E}[|\mathcal{C}|] = \Omega\left(\frac{\log \Delta}{\Delta \log \log \Delta} \cdot \alpha(G)\right).$$

If not, then $|\mathcal{C}^* \cap V_{>\Delta}| \geq \alpha(G)/2$ and thus by (1),

$$\mathbb{E}[|\mathcal{C}|] \geq \alpha(G) \cdot \left(\frac{1}{2} - \varepsilon - \frac{\varepsilon^6}{2}\right) \geq \frac{\alpha(G)}{5}.$$

We finish by plugging in the value of $\Delta$ in the first bound. $\square$

## 6. Experiments

We complement our theoretical results with an empirical evaluation for Maximum Independent Set (MIS, Section 5). The two core component to implement the algorithms in our learning-augmented setting are predictions and an approximation algorithm. In order to test our algorithms, we consider moderately-sized graphs on which we can solve MIS using a commercial integer program solver. We generate edge predictions according to the $\varepsilon$-accurate model based on these ground truth labels for varying $\varepsilon$. Then, we utilize our high/low degree MIS algorithm using the greedy MIS approximation algorithm which iteratively picks the lowest degree node which is still available.

**Hardware**   We perform all experiments on a physical workstation with an AMD Ryzen 5900X CPU and 32 GB of RAM. Deriving optimal solutions for a MIS instance requires solving an integer linear program, which is NP-hard. In practice, solvers exist that can exploit certain structures and leverage hardware to overcome the worst-case barrier for this problem. We use the CPLEX 22.1 solver (CPLEX, 2025) under an academic license.

**Datasets**   We use three graphs of varying sizes representing real social networks: *facebook* (Leskovec & Mcauley, 2012), *congress* (Fink et al., 2023), and *twitch* (Rozemberczki & Sarkar, 2021). The facebook graph has nodes representing real anonymized profiles from Facebook, with nodes representing connections between friends. It has 4039 nodes and 88234 edges; the graph has an average degree of 43.69, a median degree of 25, and a maximum degree of 1045. The congress graph represents the interactions between members of the 117th Congress on Twitter: nodes correspond to the accounts of specific members of the 117th Congress, and edges correspond to mentions of other members of Congress from a given account. This graph has 475 nodes and 13289 edges, with an maximum degree of 214,

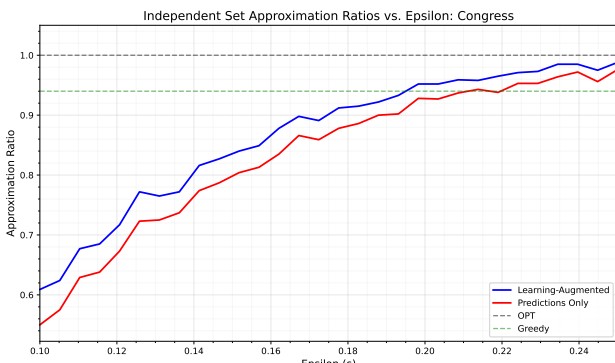

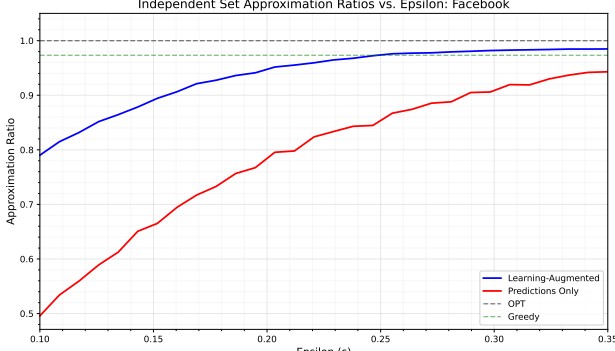

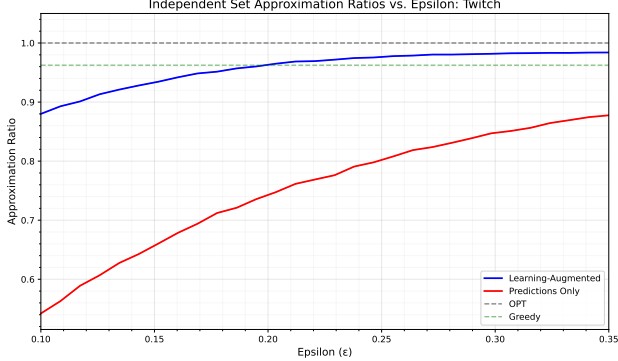

*Figure 2.* Comparison of learning-augmented frequency estimation algorithms. Top: congress, Middle: facebook, Bottom: twitch. The plots compare Algorithm 2 with the optimal solution, the standard greedy approximation of MIS, and a degree-agnostic "predictions-only" heuristic. The algorithms are averaged across 10 trials.

average degree of 43.04, and a median degree of 38. The twitch graph represents Twitch users and their shared subscriptions: nodes represent the users, and an edge between users represent whether or not the users follow a common streamer on the platform. The resulting graph is strongly connected and has 168,114 nodes and 6,797,557 edges. For the purposes of our experiment, we prune the original graph down to 50,000 nodes and around 1.1 million edges in order to make the integer linear program for finding the optimal solution feasible.

**Algorithms and Baselines** Alongside the implementation of Algorithm 2, we also include 3 baselines: the optimal solution for MIS, given by an integer linear program, a "prediction-only" heuristic, which employs the use of the edge-prediction model on the entire graph agnostic to degree, and a standard greedy algorithm approximation courtesy of Halldórsson.

**Setup** For the facebook and twitch graphs, we fix the degree threshold for Algorithm 2 to be 10, and vary $\varepsilon$ from 0.10 to 0.35. For the congress graph, we fix the degree threshold, we fix the degree threshold to 15, and vary epsilon from 0.10 to 0.25. At the upper end of these ranges of $\varepsilon$ both our algorithm and the predictions-only algorithm come close to approximation ratios of $1$. Due to the inherent randomness in making the edge predictions, we repeat each algorithm $k = 10$ times for each $\varepsilon$ and report the mean ratio.

**Results** Across varying $\varepsilon$, our algorithm outperforms the predictions-only baseline. This validates our theoretical findings where we leverage the fact that predictions are highly accurate for high degree vertices but very noisy for low degree vertices. Compared to the baseline approximation algorithm, as $\varepsilon$ (which corresponds to prediction quality) increases, we see the learning-based algorithms eventually outperforming the baseline approximation algorithm. For both datasets, we show that for an appropriate $\varepsilon$, our learning-augmented algorithm achieves the best performance with our algorithm on both graphs.

# Acknowledgements

A. Aamand was supported by the VILLUM Foundation grant 54451. J. Chen was supported by an NSF Graduate Research Fellowship under Grant No. 17453. S. Gollapudi is supported in part by the NSF (CSGrad4US award no. 2313998).

# Impact Statement

This paper presents work whose goal is to advance the field of Machine Learning. There are many potential societal consequences of our work, none which we feel must be specifically highlighted here. Our work is of theoretical nature.

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

# A. Weighted Vertex Cover

In this section, we present an algorithm for the weighted vertex cover problem, based on predictions. The main result is as follows.

**Theorem A.1.** *The set $S$ outputted by Algorithm 3 is a valid vertex cover for the input graph $G$. Furthermore,*

$$\mathbb{E}[w(S)] \leq \left(2 - \Omega\left(\frac{\log\log(1/\varepsilon)}{\log(1/\varepsilon)}\right)\right) \cdot \text{OPT}.$$

---

**Algorithm 3** Learned-WeightedVC

---

1: $S_0 \leftarrow \varnothing, \Delta \leftarrow 100\log(1/\varepsilon)/\varepsilon^2$
2: **for** $v \in V$ with $\deg(v) \geq \Delta$ **do**
3: $\quad m_v \leftarrow \text{Majority}(\{b_v(e)\}_{e \in E, v \in e})$ $\hfill \triangleright 0$ if tie
4: **end for**
5: **for** $v$ with $\deg(v) \geq \Delta$ **do**
6: $\quad N^-(v) \leftarrow \{u \in N(v) : m_u = 0 \text{ or } \deg(u) < \Delta\}$
7: $\quad$ **if** $m_v = 1$ **then**
8: $\quad\quad$ **if** $\forall u \in N(v)$: $\deg(u) \geq \Delta$ and $m_u = 1$ **then**
9: $\quad\quad\quad$ Skip $v$
10: $\quad\quad$ **else if** $w(v) < w(N^-(v))/\varepsilon^{10}$ **then**
11: $\quad\quad\quad S_0 \leftarrow S_0 \cup \{v\}$
12: $\quad\quad$ **else if** $w(v) \geq w(N^-(v))/\varepsilon^{10}$ **then**
13: $\quad\quad\quad S_0 \leftarrow S_0 \cup N^-(v)$
14: $\quad\quad$ **end if**
15: $\quad$ **else if** $m_v = 0$ **then**
16: $\quad\quad$ **if** $w(v) < \varepsilon^{10} \cdot w(N^-(v))$ **then**
17: $\quad\quad\quad S_0 \leftarrow S_0 \cup \{v\}$
18: $\quad\quad$ **else if** $w(v) \geq \varepsilon^{10} \cdot w(N^-(v))$ **then**
19: $\quad\quad\quad S_0 \leftarrow S_0 \cup N^-(v)$
20: $\quad\quad$ **end if**
21: $\quad$ **end if**
22: **end for**
23: $S_1 \leftarrow$ 2-approximate VC for heavy-heavy and heavy-light edges not covered by $S_0$
24: $S_2 \leftarrow$ a $\left(2 - 2\frac{\log\log\Delta}{\log\Delta}\right)$-approximate VC for edges not covered by $S_0 \cup S_1$
25: $S \leftarrow S_0 \cup S_1 \cup S_2$
26: **Return** $S$ $\hfill \triangleright$ Our VC approximation

---

Before we dive into the analysis, we briefly discuss the ways in which Algorithm 3 differs from Algorithm 1. The main difference is that for unweighted vertex cover, it is fine if some vertices which are not in the optimal cover are voted (by our majority vote over bits) to be in the cover we output, as long as their total *cardinality* is small. However for weighted vertex cover, this is no longer sufficient, because even a single mistake can harm us, since we may include vertices of extremely high weight. Thus, we need to perform appropriate 'sanity checks' throughout our execution and check if we can ever make a 'swap' (where we swap a vertex in our current cover with all of its uncovered neighbors). This ensures that we never add a vertex that has unreasonably high weight (formalized in the charging lemma below). One complication is that this may prevent us from adding vertices which are actually in the true optimal solution to our cover, if its (uncovered so far in the execution) neighborhood has smaller weight. However, we can also deal with this event by recognizing that this must mean one of its neighbors is misclassified, and use this event (which we show has appropriately low probability) in our charging scheme.

Recall that $\mathcal{C}$ is an optimal cover, and $\bar{\mathcal{C}} \doteq V \setminus \mathcal{C}$. Further, define $V_{\geq \Delta} \doteq \{v \in V : \deg(v) \geq \Delta\}$, and $V_{<\Delta} \doteq V \setminus V_{\geq \Delta}$. To prove Theorem A.1, we establish the following lemmas.

**Lemma A.2** (False-Positive-Weight)**.**

$$\mathbb{E}\left[w(S_0 \setminus \mathcal{C})\right] \leq \varepsilon^8 \cdot w(\mathcal{C}).$$

**Lemma A.3** (VC-Heavy-Fix-Weight). *Then*

$$\mathbb{E}\left[w(S_1)\right] \leq \varepsilon^{200} \cdot w(\mathcal{C}).$$

**Lemma A.4** (VC-Light-Weight). *Let $\mathcal{C}_{<\Delta} \doteq \mathcal{C} \setminus \mathcal{C}_{\geq\Delta}$. Then*

$$\mathbb{E}\left[w(S_2)\right] \leq \mathbb{E}\left[w(\mathcal{C}_{<\Delta} \setminus (S_0 \cup S_1)) \cdot \left(2 - 2\frac{\log\log\Delta}{\log\Delta}\right)\right].$$

Before we prove these lemmas, we apply them to finish the proof of Theorem A.1.

*Proof of Theorem A.1.* Since $S_2$ covers all edges not covered by $S_0 \cup S_1$, $S \doteq S_0 \cup S_1 \cup S_2$ is a VC. The expected cost is bounded by

$$\mathbb{E}\left[w(S)\right] \leq \mathbb{E}\left[w(S_0)\right] + \mathbb{E}\left[w(S_1)\right] + \mathbb{E}\left[w(S_2)\right].$$

$\mathbb{E}\left[w(S_1)\right]$ can be bounded by $\varepsilon^{200} \cdot w(\mathcal{C}_{\geq\Delta})$ according to Lemma A.3. The sum $\mathbb{E}\left[w(S_0)\right] + \mathbb{E}\left[w(S_2)\right]$ can be decomposed into

$$\mathbb{E}\left[w(S_0 \cap \mathcal{C})\right] + \mathbb{E}\left[w(S_0 \setminus \mathcal{C})\right] + \mathbb{E}\left[w(S_2)\right],$$

where $\mathbb{E}\left[w(S_0 \setminus \mathcal{C})\right]$ can be bounded by $\varepsilon^8 \cdot w(\mathcal{C})$ according to Lemma A.2. Finally, based on Lemma A.4,

$$\mathbb{E}\left[w(S_2)\right] \leq \mathbb{E}\left[w(\mathcal{C} \setminus S_0) \cdot \left(2 - 2\frac{\log\log\Delta}{\log\Delta}\right)\right],$$

hence

$$\mathbb{E}\left[w(S_0 \cap \mathcal{C})\right] + \mathbb{E}\left[w(S_2)\right] \leq w(\mathcal{C}) \cdot \left(2 - 2\frac{\log\log\Delta}{\log\Delta}\right).$$

Summing up the upper bounds of $\mathbb{E}\left[w(S_1)\right]$, $\mathbb{E}\left[w(S_0 \setminus \mathcal{C})\right]$ and $\mathbb{E}\left[w(S_0 \cap \mathcal{C})\right] + \mathbb{E}\left[w(S_2)\right]$ proves the lemma.

$\square$

*Proof of Lemma A.2.* To prove Lemma A.2, we need to analyze four cases where vertices not in $\mathcal{C}$ are added to $S_0$, each addressed by the following four lemmas (Lemmas A.5 to A.8). The size bound of Lemma A.2 follows directly from the sum of size bounds of the following four lemmas.

**Lemma A.5.** *Let $U_1$ be the set of vertices from $V_{\geq\Delta} \setminus \mathcal{C}$ that get added in Line 11 of Algorithm 3. We have*

$$\mathbb{E}\left[w(U_1)\right] \in O\left(\varepsilon^{100} \cdot \text{OPT}\right).$$

**Lemma A.6.** *Let $U_2$ be the set of vertices that get added in Line 13 of Algorithm 3 as a result of a vertex $v \in \mathcal{C}$. We have*

$$\mathbb{E}\left[w(U_2)\right] \in O(\varepsilon^{10} \cdot \text{OPT}).$$

**Lemma A.7.** *Let $U_3$ be the set of vertices from $\bar{\mathcal{C}}$ that we add to $S_0$ in Line 17 of Algorithm 3. We have*

$$\mathbb{E}[w(U_3)] \in O(\varepsilon^8 \cdot \text{OPT}).$$

**Lemma A.8.** *Let $U_4$ be the set of vertices from $\bar{\mathcal{C}}$ get added in Line 19 of Algorithm 3. We have*

$$\mathbb{E}[w(U_4)] \in O(\varepsilon^{100} \cdot \text{OPT}).$$

The proofs of Lemmas A.5 to A.8 are presented after those of Lemmas A.3 and A.4.

$\square$

*Proof of Lemma A.3.* To prove Lemma A.3, we need one additional lemma.

**Lemma A.9.** *Let* $U_5 \doteq C_{\geq \Delta} \setminus S_0$. *We have*

$$\mathbb{E}\left[w(U_5)\right] \in O\left(\varepsilon^{100} \cdot \text{OPT}\right). \tag{3}$$

The proof of Lemma A.9 is deferred those of Lemmas A.5 to A.8.

Now, to prove Lemma A.3, we will show that augmenting $S_0$ with an additional set $U_5$, ensures that all edges incident to at least one vertex in $V_{\geq \Delta}$ are covered. It follows that

$$\mathbb{E}\left[w(S_1)\right] \leq \mathbb{E}\left[2 \cdot w(U_5)\right] \in O(\varepsilon^{200} \cdot \text{OPT}).$$

Clearly, $S_0 \cup U_5$ covers any edge incident to any vertex in $C_{\geq \Delta}$.

It remains to consider an edge $(u, v)$ such that $v \in V_{\geq \Delta} \setminus C$ and $u \in V_{<\Delta}$. If $v$ is added to $S_0$, then $(u, v)$ is covered and we are done. Otherwise, since $u \in V_{<\Delta}$, $v$ cannot be skipped at step 9. If $v$ is not added to $S_0$, the algorithm must have added $N^-(v)$ to $S_0$. Noting that $u \in N^-(v)$ completes the proof.

$\square$

**Proof of Lemmas A.5 to A.9**

*Proof of Lemma A.4.* This simply follows from the fact that $C_{<\Delta} \setminus (S_0 \cup S_1)$, which is a subset of the optimal cover, already covers all edges not covered by $S_0 \cup S_1$. Furthermore, after removing the edges covered by $S_0 \cup S_1$, the graph has degree bounded by $\Delta$, so we can run the $\left(2 - 2\frac{\log \log \Delta}{\log \Delta}\right)$ approximation algorithm on this graph from (Halperin, 2002). $\square$

*Proof of Lemma A.5.* First, since $v \in \bar{C}$, all its neighbors must belong to $C$.

We charge the cost $w(v)$ of selecting $v \in V_{\geq \Delta} \setminus C$ in Line 11 of Algorithm 3 to its neighbors in $N^-(v)$.

For each $u \in N^-(v)$, we charge at most $w(u)/\varepsilon^{10}$, ensuring that the total charge is at most $w(N^-(v))/\varepsilon^{10}$, which covers $w(v)$. Define an indicator $X_{v \to u}$ for this event. If $u \in C_{<\Delta}$, Lemma E.1 gives:

$$\Pr\left[X_{v \to u} = 1\right] \leq \Pr\left[m_v = 1\right]$$
$$= \exp\left(-2 \cdot \deg(v) \cdot \varepsilon^2\right) \leq \exp\left(-2 \cdot \Delta \cdot \varepsilon^2\right).$$

If $u \in C_{\geq \Delta}$, then $u \in N^-(v)$ implies $m_u = 0$, yielding:

$$\Pr\left[X_{v \to u} = 1\right] \leq \Pr\left[m_u = 0\right] = \exp\left(-2 \cdot \deg(u) \cdot \varepsilon^2\right).$$

Set $X_{v \to u} \doteq 0$ in all other cases.

Now, for each $u \in C$, define $X_u \doteq \sum_{v \in N(u)} X_{v \to u}$. Thus, we bound $w(U_1)$ by:

$$\sum_{u \in C_{<\Delta}} \frac{w(u)}{\varepsilon^{10}} \cdot X_u + \sum_{u \in C_{\geq \Delta}} \frac{w(u)}{\varepsilon^{10}} \cdot X_u.$$

For each $u \in C_{<\Delta}$, since $\Delta = 100 \log(1/\varepsilon)/\varepsilon^2$,

$$\mathbb{E}\left[\frac{w(u)}{\varepsilon^{10}} \cdot X_u\right] \leq \frac{w(u)}{\varepsilon^{10}} \cdot \Delta \cdot \exp\left(-2 \cdot \Delta \cdot \varepsilon^2\right) \leq w(u) \cdot \varepsilon^{100}.$$

For each $u \in C_{\geq \Delta}$,

$$\mathbb{E}\left[\frac{w(u)}{\varepsilon^{10}} \cdot X_u\right] \leq \frac{w(u)}{\varepsilon^{10}} \cdot \deg(u) \cdot \exp\left(-2 \cdot \deg(u) \cdot \varepsilon^2\right) \leq w(u) \cdot \varepsilon^{100}.$$

Therefore,

$$\mathbb{E}\left[w(U_1)\right] \leq \mathbb{E}\left[\sum_{u \in \mathcal{C}_{<\Delta}} \frac{w(u)}{\varepsilon^{10}} \cdot X_u + \sum_{u \in \mathcal{C}_{\geq\Delta}} \frac{w(u)}{\varepsilon^{10}} \cdot X_u\right] \in O(\varepsilon^{100} \cdot \text{OPT}).$$

$\square$

*Proof of Lemma A.6.* Every time we add $N^-(v)$ in Line 13, the total weight of this set is at most $\varepsilon^{10} w(v)$. We can do this at most once per vertex $v \in \mathcal{C}$, so the lemma follows. $\square$

*Proof of Lemma A.7.* First, since $v \in \bar{\mathcal{C}}$, all its neighbors must belong to $\mathcal{C}$.

We charge the cost $w(v)$ of selecting $v \in V_{\geq\Delta} \setminus \mathcal{C}$ in Line 17 of Algorithm 3 to its neighbors in $N^-(v)$.

For each $u \in N^-(v)$, we charge at most $\varepsilon^{10} \cdot w(u)$, ensuring that the total charge is at most $\varepsilon^{10} \cdot w(N^-(v))$, which covers $w(v)$. Define an indicator $X_{v \to u}$ for this event. If $u \in \mathcal{C}_{<\Delta}$,

$$\Pr\left[X_{v \to u} = 1\right] \leq 1.$$

If $u \in \mathcal{C}_{\geq\Delta}$, then $u \in N^-(v)$ implies $m_u = 0$, yielding:

$$\Pr\left[X_{v \to u} = 1\right] \leq \Pr\left[m_u = 0\right] = \exp\left(-2 \cdot \deg(u) \cdot \varepsilon^2\right).$$

Set $X_{v \to u} \doteq 0$ in all other cases.

Now, for each $u \in \mathcal{C}$, define $X_u \doteq \sum_{v \in N(u)} X_{v \to u}$. Thus, we bound $w(U_1)$ by:

$$\sum_{u \in \mathcal{C}_{<\Delta}} \varepsilon^{10} \cdot w(u) \cdot X_u + \sum_{u \in \mathcal{C}_{\geq\Delta}} \varepsilon^{10} \cdot w(u) \cdot X_u.$$

For each $u \in \mathcal{C}_{<\Delta}$, since $\Delta = 100 \log(1/\varepsilon)/\varepsilon^2$,

$$\mathbb{E}\left[\varepsilon^{10} \cdot w(u) \cdot X_u\right] \leq \varepsilon^{10} \cdot w(u) \cdot \Delta \leq 100 \cdot \varepsilon^8 \cdot w(u).$$

For each $u \in \mathcal{C}_{\geq\Delta}$,

$$\mathbb{E}\left[\varepsilon^{10} \cdot w(u) \cdot X_u\right] \leq \varepsilon^{10} \cdot w(u) \cdot \deg(u) \cdot \exp\left(-2 \cdot \deg(u) \cdot \varepsilon^2\right) \leq w(u) \cdot \varepsilon^{110}.$$

Therefore,

$$\mathbb{E}\left[w(U_1)\right] \leq \mathbb{E}\left[\sum_{u \in \mathcal{C}_{<\Delta}} \varepsilon^{10} \cdot w(u) \cdot X_u + \sum_{u \in \mathcal{C}_{\geq\Delta}} \varepsilon^{10} \cdot w(u) \cdot X_u\right] \in O(\varepsilon^8 \cdot \text{OPT}).$$

$\square$

*Proof of Lemma A.8.* Let $v \in \mathcal{C}_{\geq\Delta}$. Since $\deg(v) \geq \Delta = 100 \log(1/\varepsilon)/\varepsilon^2$, The probability of it being misclassified is at most $\varepsilon^{200}$ from Lemma E.1. Furthermore, we only add $N^-(v)$ in Line 19 if $w(v) \geq \varepsilon^{10} \cdot N^-(v)$. Thus, if we add $N^-(v)$, then we add weight at most $w(v)/\varepsilon^{10}$. Altogether, this means that

$$\mathbb{E}[w(U_4)] \leq \sum_{v \in \mathcal{C}_{\geq\Delta}} \Pr(v \text{ is misclassified}) \cdot \frac{w(v)}{\varepsilon^{10}} \leq O(\varepsilon^{100} \cdot \text{OPT}).$$

$\square$

*Proof of Lemma A.9.* A $v \in \mathcal{C}_{\geq \Delta}$ can be skipped at step 9, step 13 or step 19.

First, consider a vertex $v \in \mathcal{C}_{\geq \Delta}$ is skipped at step 9. For each vertex $v \in \mathcal{C}$, it has at least one neighbor in $\bar{\mathcal{C}}$. Denote this neighbor as $u$. Then we know $\deg(u) \geq \Delta$ and $m_u = 1$. Therefore, based on Lemma E.1, and given that $\deg(u) \geq \Delta = 100/\varepsilon^2 \cdot \log(1/\varepsilon)$, we obtain

$$\Pr\left[v \text{ is skipped at step 9}\right] \leq \Pr\left[m_u = 1\right] \leq \exp\left(-2 \cdot \deg(u) \cdot \varepsilon^2\right) \leq \varepsilon^{200}. \tag{4}$$

Next, consider $v \in \mathcal{C}_{\geq \Delta}$ is skipped at step 12. Since $v \in \mathcal{C}$, it holds that

$$w(v) \leq \sum_{u \in N(v) \setminus \mathcal{C}} w(u),$$

as otherwise $(\mathcal{C} \cup (N(v) \setminus \mathcal{C})) \setminus \{v\}$ is a better vertex cover than $\mathcal{C}$, a contradiction. For each $u \in N(v) \setminus \mathcal{C}$, define the indicator $Y_u$ for $u$ being in $N^-(v)$. Note that if $\deg(u) < \Delta$, then we must have $Y_u = 1$ and $\mathbb{V}\mathrm{ar}\left[Y_u\right] = 0$. If $\deg(u) \geq \Delta$, then based on Lemma E.1,

$$\Pr\left[Y_u = 1\right] = \Pr\left[m_u = 0\right] \geq 1 - \exp(-2\varepsilon^2 \deg(u)) \geq 1 - \varepsilon^{200}, \quad \mathbb{V}\mathrm{ar}\left[Y_u\right] \leq \varepsilon^{200}. \tag{5}$$

Further,

$$\mathbb{E}\left[\sum_{u \in N(v) \setminus \mathcal{C}} w(u) \cdot Y_u\right] \geq (1 - \varepsilon^{200}) \cdot \sum_{u \in N(v) \setminus \mathcal{C}} w(u) \geq (1 - \varepsilon^{200}) \cdot w(v).$$

Now, $v$ is skipped at step 12, which implies that

$$w(v) \geq \frac{w(N^-(v))}{\varepsilon^{10}} \geq \frac{1}{\varepsilon^{10}} \sum_{u \in N(v) \setminus \mathcal{C}} w(u) \cdot Y_u. \tag{6}$$

Therefore,

$$\Pr\left[\sum_{u \in N(v) \setminus \mathcal{C}} w(u) \cdot Y_u \leq \varepsilon^{10} \cdot w(v)\right] \tag{7}$$

$$\leq \Pr\left[\left|\sum_{u \in N(v) \setminus \mathcal{C}} w(u) \cdot Y_u - \mathbb{E}\left[\sum_{u \in N(v) \setminus \mathcal{C}} w(u) \cdot Y_u\right]\right| \geq (1 - \varepsilon^{200} - \varepsilon^{10}) \cdot w(v)\right] \tag{8}$$

$$\leq \frac{\sum_{u \in N(v) \setminus \mathcal{C}} w(u)^2 \cdot \mathbb{V}\mathrm{ar}\left[Y_u\right]}{((1 - \varepsilon^{200} - \varepsilon^{10}) \cdot w(v))^2} \tag{9}$$

$$\leq \frac{\varepsilon^{200} \left(\sum_{u \in N(v) \setminus \mathcal{C}} w(u)\right)^2}{((1 - \varepsilon^{200} - \varepsilon^{10}) \cdot w(v))^2} \tag{10}$$

$$\leq \varepsilon^{100}. \tag{11}$$

Finally, consider a vertex $v \in \mathcal{C}_{\geq \Delta}$ is skipped at step 19. This can be bounded by the probability $\Pr\left[m_v = 0\right] \leq \varepsilon^{200}$.

Thus,

$$\mathbb{E}\left[w(U_5)\right] \leq \varepsilon^{100} \sum_{v \in \mathcal{C}_{\geq \Delta}} w(v) \leq \varepsilon^{100} \cdot \mathrm{OPT}. \tag{12}$$

$\square$

## B. Set Cover

In this section, we present Algorithm 4 for SC, achieving an approximation ratio better than $\ln m$. We first extend the edge-based prediction model from Definition 1.1 to SC. Recall from Section 2 that in the SC problem, we treat subsets as "vertices" and elements as "edges".

**Definition B.1** (Prediction Model)**.** Given a family of subsets $S_1, \ldots, S_n \subseteq [m]$, let $\mathcal{J}^* \subseteq [n]$ be a fixed optimal set cover. Each element $i \in [m]$ outputs a bit $b_j(i)$ for every subset $S_j$ containing $i$. If $j \in \mathcal{J}^*$, then $b_j(i) = 1$ with probability $1/2 + \varepsilon$ and 0 otherwise. All bits are independent across predictions.

The main result is as follows. Recall that approximating the Set Cover problem (without prediction) within a factor of $(1 - \delta) \ln m$ is NP-hard for any $\delta > 0$ (Dinur & Steurer, 2014).

**Theorem B.2** (Learned Set Cover)**.** *Let $\Delta \doteq 100/\varepsilon^2 \ln(1/\varepsilon)$, and $\mathcal{J}^*$ be an optimal solution. Algorithm 4 returns a solution $J$ satisfying*

$$\mathbb{E}[|J|] \leq \left(1 + \varepsilon^{10} + \ln \Delta\right) \cdot |\mathcal{J}^*|. \tag{13}$$

Observe that, for a given $\varepsilon$, $(1 + \varepsilon^{10} + \ln \Delta)$ is a constant which does not depend on $m$.

**Algorithm**   Algorithm 4 follows a similar philosophy to Algorithm 1. It first determines whether to include subsets of size at least $\Delta$ in the solution based on predicted information. However, unlike Algorithm 1, it simply ignores large subsets when predictions suggest they are not part of the optimal solution.

After selecting large subsets, Algorithm 4 applies the $(1 + \ln \Delta)$-approximation algorithm (Lovász, 1975) to small subsets, aiming to cover as many uncovered elements as possible. Finally, it covers any remaining elements using an $\ln m$-approximation algorithm (Lovász, 1975) with all available subsets.

---

**Algorithm 4** Learned Set Cover

1: $J_{learned} \leftarrow \varnothing, \Delta \leftarrow 100 \log(1/\varepsilon)/\varepsilon^2$
2: Sort the $S_j$'s in decreasing order according to $|S_j|$
3: **for** $j \in [n]$ with $|S_j| \geq \Delta$ **do**
4:      $b_j \leftarrow \text{Majority}(\{b_j(i)\}_{i \in S_j})$                                                  ▷ 0 if tie
5:      **if** $b_j = 1$ and $S_j \not\subseteq \cup_{j' \in J_{learned}} S_{j'}$ **then**
6:          $J_{learned} \leftarrow J_{learned} \cup \{j\}$
7:      **end if**
8: **end for**
9: $U_{learned} \leftarrow \cup_{j \in J_{learned}} S_j$
10: $\mathcal{I}_{<\Delta} \leftarrow \{j \in [n] : |S_j| < \Delta\}$
11: $U_{approx} \leftarrow (\cup_{j \in \mathcal{I}_{<\Delta}} S_j) \setminus U_{learned}$
12: $J_{approx} \leftarrow$ a $(1 + \log \Delta)$-approximate Set Cover on $(U_{approx}, \mathcal{I}_{<\Delta})$
13: $J_{fix} \leftarrow$ a $(\ln m)$-approximate Set Cover on $([m] \setminus (U_{learned} \cup U_{approx}), [n])$
14: **return** $J_{learned} \cup J_{approx} \cup J_{fix}$

---

**Analysis**   The proof of Theorem B.2 relies on the following lemmas. At a high level, the proofs of the lemmas involve bounding the expected number of optimal subsets not selected by Algorithm 4 due to prediction errors (False Negatives), and applying a "charging argument" to bound the expected number of non-optimal subsets incorrectly selected (False Positives).

**Lemma B.3** (False Positive)**.** *Define $\mathcal{J}^*_{\geq \Delta} \doteq \{j \in \mathcal{J}^* : |S_j| \geq \Delta\}$, and $J_{FP} \doteq J_{learned} \setminus \mathcal{J}^*_{\geq \Delta}$.*

$$\mathbb{E}[|J_{FP}|] \leq \varepsilon^{10} \cdot |\mathcal{J}^*|. \tag{14}$$

**Lemma B.4** (False Negative Neighbors)**.** *Define $\mathcal{J}^*_{\geq \Delta} \doteq \{j \in \mathcal{J}^* : |S_j| \geq \Delta\}$, and $J_{FN} \doteq \mathcal{J}^*_{\geq \Delta} \setminus J_{learned}$. Then*

$$\mathbb{E}\left[\sum_{j \in J_{FN}} |S_j|\right] \leq \varepsilon^{10} \cdot \left|\mathcal{J}^*_{\geq \Delta}\right|.$$

**Lemma B.5.** *Let $\mathcal{J}^*_{approx}$ be the optimal set cover for $(U_{approx}, \mathcal{I}_{<\Delta})$. Then*

$$\mathbb{E}\left[\left|\mathcal{J}^*_{approx}\right|\right] \leq \mathcal{J}^*_{<\Delta} + \varepsilon^{10} \cdot \left|\mathcal{J}^*_{\geq\Delta}\right|. \tag{15}$$

**Lemma B.6.** *Define $U_{fix} \doteq [m] \setminus \cup_{j \in J_{learned} \cup J_{approx}} S_j$.*

$$\mathbb{E}\left[|U_{fix}|\right] \leq \varepsilon^{10} \cdot \left|\mathcal{J}^*_{\geq\Delta}\right|. \tag{16}$$

Before we prove Lemmas B.3 to B.6, we apply them to finish the proof of our main result Theorem B.2.

*Proof of Theorem B.2.* Observe that

$$|J| \leq |J_{learned}| + |J_{approx}| + |J_{fix}| \leq \left|\mathcal{J}^*_{\geq\Delta}\right| + \left|J_{learned} \setminus \mathcal{J}^*_{\geq\Delta}\right| + |J_{approx}| + |J_{fix}|.$$

Bounding the second term with Lemma B.4, the third term with Lemma B.5 and that $J_{approx}$ being an $(1 + \log \Delta)$ approximate solution, and the last term with Lemma B.6 gives

$$|J| \leq \left|\mathcal{J}^*_{\geq\Delta}\right| + \varepsilon^{10} \cdot |\mathcal{J}^*| + (1 + \ln\Delta)\left(\mathcal{J}^*_{<\Delta} + \varepsilon^{10} \cdot \left|\mathcal{J}^*_{\geq\Delta}\right|\right) + \varepsilon^{10} \cdot \left|\mathcal{J}^*_{\geq\Delta}\right| \leq \left(1 + \varepsilon^{10} + \ln\Delta\right) \cdot |\mathcal{J}^*|.$$

$\square$

**Proof of Lemmas B.3 to B.6**

In order to prove these four lemmas, we require the following technical result.

**Lemma B.7** (Incorrect predictions)**.** *For each $j \in [n]$ s.t., $|S_j| \geq \Delta$, it holds that*

$$\Pr\left[\mathbb{1}_{[j \in J_{learned}]} \neq \mathbb{1}_{[j \in \mathcal{J}^*]}\right] \leq 2 \cdot \exp\left(-2 \cdot |S_j| \cdot \varepsilon^2\right). \tag{17}$$

*Proof of Lemma B.7.* For each $i \in S_j$, let $X_i \in \{0, 1\}$ be the prediction by element $i$ of the event $j \in \mathcal{J}^*$.

We consider two cases. If $j \notin \mathcal{J}^*$, then $b_j \neq \mathbb{1}_{[j \in \mathcal{J}^*]}$ if $\sum_{i \in S_j} X_i > |S_j|/2$. Note that $\mathbb{E}\left[\sum_{i \in S_j} X_i\right] = |S_j| \cdot (1/2 - \varepsilon)$. By Hoeffding's inequality, when $|S_j| \geq \Delta$:

$$\Pr\left[b_j \neq \mathbb{1}_{[j \in \mathcal{J}^*]}\right] = \Pr\left[\sum_{i \in S_j} X_i > |S_j|/2\right] \leq \exp\left(-2 \cdot \frac{|S_j|^2 \cdot \varepsilon^2}{|S_j|}\right) = \exp\left(-2 \cdot |S_j| \cdot \varepsilon^2\right).$$

Therefore,

$$\Pr\left[\mathbb{1}_{[j \in J_{learned}]} \neq \mathbb{1}_{[j \in \mathcal{J}^*]}\right] = \Pr\left[j \in J_{learned}\right] \leq \Pr\left[b_j = 1\right] \leq \exp\left(-2 \cdot |S_j| \cdot \varepsilon^2\right). \tag{18}$$

If $j \in \mathcal{J}^*$, then $b_j \neq \mathbb{1}_{[j \in \mathcal{J}^*]}$ if $\sum_{i \in S_j} X_i \leq |S_j|/2$. Note that $\mathbb{E}\left[\sum_{i \in S_j} X_i\right] = |S_j| \cdot (1/2 + \varepsilon)$. By Hoeffding's inequality, when $|S_j| \geq \Delta$:

$$\Pr\left[b_j \neq \mathbb{1}_{[j \in \mathcal{J}^*]}\right] = \Pr\left[\sum_{i \in S_j} X_i \leq |S_j|/2\right] \leq \exp\left(-2 \cdot \frac{|S_j|^2 \cdot \varepsilon^2}{|S_j|}\right) = \exp\left(-2 \cdot |S_j| \cdot \varepsilon^2\right).$$

Observe that

$$\Pr\left[\mathbb{1}_{[j \in J_{learned}]} \neq \mathbb{1}_{[j \in \mathcal{J}^*]}\right] = \Pr\left[j \notin J_{learned}\right] \tag{19}$$
$$= \Pr\left[j \notin J_{learned} \wedge b_j = 0\right] + \Pr\left[j \notin J_{learned} \wedge b_j = 1\right]. \tag{20}$$

The former probability is bounded by

$$\Pr\left[j \notin J_{learned} \wedge b_j = 0\right] = \Pr\left[b_j = 0\right] \leq \exp\left(-2 \cdot |S_j| \cdot \varepsilon^2\right).$$

When $b_j = 1$, the algorithm decides not to add $j$ to $J_{learned}$ (therefore $j \notin J_{learned}$) if $S_j \subseteq \bigcup_{j' \in J_{learned}} S_{j'}$. However, for the optimal cover $\mathcal{J}^*$, it holds that $S_j \not\subseteq \cup_{j' \in \mathcal{J}^* \setminus \{j\}} S_{j'}$, as otherwise we can remove $j$ from $\mathcal{J}^*$. Therefore, $J_{learned} \setminus \mathcal{J}^* \neq \varnothing$. Let $j'$ be some index from $J_{learned} \setminus \mathcal{J}^*$. It is added to $J_{learned}$ only if $b_{j'} = 1$. As $j'$ is processed before $j$, we have $|S_{j'}| \geq |S_j|$. It concludes that

$$\Pr\left[j \notin J_{learned} \wedge b_j = 1\right] \leq \Pr\left[b_{j'} = 1\right] \leq \exp\left(-2 \cdot |S_{j'}| \cdot \varepsilon^2\right) \leq \Pr\left[b_j = 1\right] \leq \exp\left(-2 \cdot |S_j| \cdot \varepsilon^2\right).$$

□

We can now provide the proofs of Lemmas B.3 to B.6.

*Proof of Lemma B.3.* Define $\mathcal{B} \doteq \{j \in [n] \setminus \mathcal{J}^* : |S_j| \geq \Delta\}$. We are going to charge the cost of selecting sets from $\mathcal{B}$, to sets in $\mathcal{J}^*$, as follows: Consider a fix $j \in \mathcal{B}$ and assume that $j$ is selected into $J_{learned}$. Since $j$ is selected only if $S_j \not\subseteq \bigcup_{j' \in J_{learned}} S_{j'}$ (at the time $j$ is selected), there exists $i \in S_j \setminus \left(\bigcup_{j' \in J_{learned}} S_{j'}\right)$.

Further, there exists some $S_{j'}$ for $j' \in \mathcal{J}^*$ which covers element $i$. It is easy to see $j' \notin \bigcup_{j' \in J_{learned}} S_{j'}$ when $j$ is selected. We will charge the cost selecting set $S_j$ to $S_{j'}$, by setting $X_{j,j'} = 1$. Since Algorithm 4 selects sets into $J_{learned}$ in decreasing order according to their sizes, either of one the following happens:

1. $|S_{j'}| \leq |S_j|$, then $\Pr\left[X_{j,j'} = 1\right] \leq \Pr\left[b_j = 1\right] \leq \exp\left(-2 \cdot |S_j| \cdot \varepsilon^2\right) \leq \exp\left(-2 \cdot \Delta \cdot \varepsilon^2\right)$.

2. $|S_{j'}| > |S_j|$ and $j' \notin J_{learned}$, then $\Pr\left[X_{j,j'} = 1\right] \leq \Pr\left[j' \notin J_{learned}\right] \leq \exp\left(-2 \cdot |S_{j'}| \cdot \varepsilon^2\right)$.

To make $X_{j,j'}$ well defined, we set $X_{j,j'} = 0$ for all other cases. Combining both cases, we have

$$\Pr\left[X_{j,j'} = 1\right] \leq \exp\left(-2 \cdot \max\left\{|S_j|, |S_{j'}|, \Delta\right\} \cdot \varepsilon^2\right). \tag{21}$$

Finally, since,

$$|J_{FP}| = \sum_{j \in \mathcal{B}, j' \in \mathcal{J}^*} X_{j,j'} = \sum_{j' \in \mathcal{J}^*} \left(\sum_{j \in \mathcal{B}} X_{j,j'}\right).$$

we conclude that

$$\mathbb{E}\left[|J_{FP}|\right] \leq \sum_{j' \in \mathcal{J}^*} |S_{j'}| \cdot \exp\left(-2 \cdot \max\left\{|S_j|, |S_{j'}|, \Delta\right\} \cdot \varepsilon^2\right) \leq |\mathcal{J}^*| \cdot \varepsilon^{10}. \tag{22}$$

□

*Proof of Lemma B.4.* Consider a fixed $j \in \mathcal{J}^*_{\geq \Delta}$. Via Lemma B.7, it holds that

$$\mathbb{E}[\mathbb{1}_{[j \in J_{FN}]} \cdot |S_j|] = |S_j| \cdot \Pr\left[\mathbb{1}_{[j \in J_{learned}]} \neq \mathbb{1}_{[j \in \mathcal{J}^*]}\right] \leq |S_j| \cdot \exp\left(-2 \cdot |S_j| \cdot \varepsilon^2\right) \leq \Delta \cdot \exp\left(-2 \cdot \Delta \cdot \varepsilon^2\right),$$

where the final inequality holds since $y = -2 \cdot x \cdot \varepsilon^2 + \ln x$ decreases when $x \geq 1/(2 \cdot \varepsilon^2)$, and $|S_j| \geq \Delta \geq 1/(2 \cdot \varepsilon^2)$. It follows that

$$\mathbb{E}\left[\sum_{j \in J_{FN}} |S_j|\right] = \mathbb{E}\left[\sum_{j \in \mathcal{J}^*_{\geq \Delta}} \mathbb{1}_{[j \in J_{FN}]} \cdot |S_j|\right] \leq |\mathcal{J}^*_{\geq \Delta}| \cdot \Delta \cdot \exp\left(-2 \cdot \Delta \cdot \varepsilon^2\right) \leq |\mathcal{J}^*_{\geq \Delta}| \cdot \varepsilon^{10}, \tag{23}$$

where the final inequality holds since $\Delta \geq 100/\varepsilon^2 \cdot \log(1/\varepsilon)$. □

*Proof of Lemma B.5.* Define $U^*_{\geq \Delta} \doteq \bigcup_{j \in \mathcal{J}^*_{\geq \Delta}} S_j$, $U^*_{< \Delta} \doteq \bigcup_{j \in \mathcal{J}^*_{< \Delta}} S_j$. Then $U^*_{\geq \Delta}$ and $U^*_{< \Delta}$ is a partition of $[m]$. Since $\mathcal{J}^*_{\geq \Delta} \subseteq J_{learned} \cup J_{FN}$, it holds that

$$[m] \setminus \bigcup_{j \in J_{learned}} S_j \subseteq U^*_{< \Delta} \cup (\cup_{j \in J_{FN}} S_j). \tag{24}$$

Since $\mathcal{J}_{<\Delta}^*$ covers $U_{<\Delta}^*$, and $\cup_{j \in J_{FN}} S_j$ can be covered by at most $\sum_{j \in J_{FN}} |S_j|$ sets,

$$\left|\mathcal{J}_{approx}^*\right| \leq \left|\mathcal{J}_{<\Delta}^*\right| + \sum_{j \in J_{FN}} |S_j|. \tag{25}$$

Taking expectation of both sides and applying Lemma B.4 finish the proof. $\square$

*Proof of Lemma B.6.* Since
$$U_{fix} = [m] \setminus (U_{learned} \cup U_{approx}) \subseteq \cup_{j \in J_{FN}} S_j, \tag{26}$$

we have
$$|U_{fix}| \leq \sum_{j \in J_{FN}} |S_j|. \tag{27}$$

Taking expectation of both sides and applying Lemma B.4 finish the proof. $\square$

# C. Maximum Cut

Let $G = (V, E)$ be a undirected and weighted graph without self loops, and $A \in \mathbb{R}_{\geq 0}^{n \times n}$ be a symmetric adjacent matrix, where $A_{i,j} = w_{i,j} \geq 0$, the weight of edge $(i, j)$ if it exists, and $0$ otherwise. The MAXCUT problem looks for the cut with maximum edge weight, and can be formulated as the following maximization problem

$$\max_{x \in \{-1,1\}^n} \ (1/4) \cdot \sum_{i,j \in [n]} w_{i,j} \cdot (x_i - x_j)^2. \tag{28}$$

**Definition C.1** (Prediction Model). Given an input weight and undirected graph $G$, we fix some optimal solution $b^* = (b_1^*, \ldots, b_n^*) \in \{-1, 1\}^n$. Every edge $e = (i, j)$ outputs two bits $(b_i(e), b_j(e))$ of predictions. $b_i(e) = b_i^*$ with probability $1/2 + \varepsilon$ and $-b_i^*$ otherwise. $b_j(e)$ is similarly sampled and all predictions are i.i.d. across all edges and bits.

Our main result is stated as follows.

**Theorem C.2.** *There is a randomized algorithm for the* MAXCUT *problem under our prediction model which achieve approximation ratio of* $\alpha_{GW} + \tilde{\Omega}(\varepsilon^2)$.

## C.1. Algorithm

We present the road-map of the proof for the Theorem C.2. Let $\Delta \in \mathbb{N}^+, \eta \in (0, 1)$ be fixed parameters to be set later.

**Definition C.3.** For each vertex $i \in [n]$, the $\Delta$-prefix for $i$ comprises the $\Delta$ heaviest edges incident to $i$ (breaking ties arbitrarily), while the $\Delta$-suffix comprises the remaining edges.

**Definition C.4** (($\Delta, \eta$)-Narrow/Wide Vertex). A vertex $i$ is $(\Delta, \eta)$-wide if the total weight of edges in its $\Delta$-prefix is at most $\eta \cdot W_i$, where $W_i \doteq \sum_{j \in N(i)} A_{ij}$ is the weighted degree of $i$. Otherwise, the vertex $i$ is $(\Delta, \eta)$-narrow.

Intuitively, a $(\Delta, \eta)$-wide vertex is one where the weights of the edges incident to $i$ are evenly distributed.

**Definition C.5** (($\Delta, \eta$)-Narrow/Wide Graph). A graph is $(\Delta, \eta)$-wide if the sum of weighted degrees of $(\Delta, \eta)$-narrow vertices is at most $\eta \cdot W$, where $W \doteq \sum_{i \in [n]} W_i$. Otherwise, the graph $i$ is $(\Delta, \eta)$-narrow.

$(\Delta, \eta)$**-Narrow Graph.** We apply the following result from (Cohen-Addad et al., 2024), which does not rely on the predicted information.

**Proposition C.6** ((Cohen-Addad et al., 2024)). *Given a* $(\Delta, \eta)$-*narrow graph, there is a randomized algorithm for the* MAXCUT *problem with an (expected) approximation ratio of* $\alpha_{GW} + \tilde{\Omega}(\eta^5/\Delta^2)$.

$(\Delta, \eta)$**-Wide Graph.** We are going to prove the following key result. Algorithm 5 is adapted from (Ghoshal et al., 2025). The original work is based on the vertex prediction model and does not include a proof for $(\Delta, \eta)$-wide graphs. We extend their proof to cover this case.

**Theorem C.7.** *Given a* $(\Delta, \eta)$-*wide graph and the prediction model of Definition C.1, there is a randomized algorithm (Algorithm 5) for the* MAXCUT *problem with an (expected) approximation ratio of*

$$1 - O\left(\frac{\sqrt{\eta}}{\varepsilon \cdot \Delta} + \eta\right) \tag{29}$$

*Proof of Theorem C.2.* Combining Proposition C.6 and Theorem C.7, and by setting $\eta$ to be suitably small universal constant, and setting $\Delta = 1/(c \cdot \varepsilon)$ for some suitably small universal constant $c$ finish the proof. $\square$

*Remark* C.8. The paper (Cohen-Addad et al., 2024) achieves a ration of

$$0.98\left(1 - O\left(\frac{1}{\varepsilon \cdot \sqrt{\Delta}} + \eta\right)\right) + 0.02 \cdot \alpha_{GW} \tag{30}$$

for $(\Delta, \eta)$-wide graphs. Therefore, they need to set $\Delta = 1/(c\varepsilon^2)$, resulting in a final ratio of $\alpha_{GW} + \tilde{\Omega}(\varepsilon^4)$ for $(\Delta, \eta)$-narrow graphs.

The proof of Theorem C.7 relies on the following lemmas.

---

**Algorithm 5** Learning Based MAXCUT

---

1: **Truncation.** Define the matrix $\tilde{A}_{i,j}$ by

$$\tilde{A}_{i,j} = \begin{cases} 0, & \forall j \in [n], \text{ if } i \text{ is } (\Delta, \eta)\text{-narrow} \\ \min\{A_{i,j},\, \eta \cdot W_i/\Delta\}, & \forall j \in [n], \text{ if } i \text{ is } (\Delta, \eta)\text{-wide} \end{cases} \tag{31}$$

2: **Prediction.**

$$\forall i \in [n], \quad z_i \doteq \frac{1}{\deg(i)} \cdot \sum_{e \in E, i \in e} \frac{b_i(e)}{2\varepsilon}. \tag{32}$$

3: **Optimization.** Solve the convex optimization problem

$$\min_{x \in [-1,1]^n} x^T \tilde{A} z + \left\| \tilde{A} z - \tilde{A} x \right\|_1 \tag{33}$$

4: **Rounding.** For each $i \in [n]$, choose $y_i \in \{-1, 1\}$, which minimizes

$$(y_{1:i-1}, y_i, x_{i+1:n})^T \tilde{A} (y_{1:i-1}, y_i, x_{i+1:n}), \tag{34}$$

where $(y_{1:i-1}, y_i, x_{i+1:n})$ is the vector whose first $i-1$ entries comprise the chosen $y_1, \ldots, y_{i-1}$, the $i^{(th)}$ entry is $y_i$, and the remaining entries are $x_{i+1}, \ldots, x_n$.

---

**Lemma C.9.** *and let* $\mathcal{E}rr \doteq \sum_{i \in [n]} \sqrt{\sum_{j \in [n]} \tilde{A}_{i,j}^2 \cdot \mathbb{V}\mathrm{ar}\,[z_j]}$. *Algorithm 5 returns a solution* $y'$ *satisfying*

$$\mathbb{E}_z \left[ y^T A y - (b^*)^T A b^* \right] \le 2\eta W + \mathcal{E}rr. \tag{35}$$

**Lemma C.10.** *Given prediction model in Definition C.1, and an* $(\Delta, \eta)$-*wide graph, it holds that*

$$\mathcal{E}rr \le \frac{\sqrt{\eta}}{\varepsilon \cdot \Delta} \cdot W \tag{36}$$

*Proof of Theorem C.7.* First, since $(b_i^*)^2 = y_i^2 = 1$, we have

$$\begin{aligned}
\sum_{i \ne j} w_{i,j}(b_i^* - b_j^*)^2 &= \sum_{i \ne j} w_{i,j}\left( (b_i^*)^2 + (b_j^*)^2 - 2 \cdot b_i^* \cdot b_j^* \right) \\
&= \sum_{i \ne j} w_{i,j}\left( y_i^2 + y_j^2 - 2 \cdot b_i^* \cdot b_j^* \right) \\
&= \sum_{i \ne j} w_{i,j}\left( y_i^2 + y_j^2 - 2 \cdot y_i \cdot y_j + 2 \cdot y_i \cdot y_j - 2 \cdot b_i^* \cdot b_j^* \right) \\
&= \sum_{i \ne j} w_{i,j}(y_i - y_j)^2 + 2 \cdot y^T A y - 2 \cdot (b^*)^T A b^*.
\end{aligned}$$

Therefore,

$$\frac{1}{4} \cdot \sum_{i \ne j} w_{i,j}(y_i - y_j)^2 = \frac{1}{4} \sum_{i \ne j} w_{i,j}(b_i^* - b_j^*)^2 - \frac{1}{2} \cdot \left( y^T A y - (b^*)^T A b^* \right) = opt - \frac{1}{2} \cdot \left( y^T A y - (b^*)^T A b^* \right).$$

Combing Lemmas C.9 and C.10 and noting that $opt \ge W/2$ gives

$$y^T A y - (b^*)^T A b^* \in O\left( \left( \frac{\sqrt{\eta}}{\varepsilon \cdot \Delta} + \eta \right) \cdot opt \right), \tag{37}$$

which proves the claim. $\qquad\square$

To prove Lemma C.9, we need the following lemmas.

**Lemma C.11.**

$$\mathbb{E}\left[ \left\| \tilde{A} b^* - \tilde{A} z \right\|_1 \right] \le \mathcal{E}rr \tag{38}$$

**Lemma C.12** (Rounding Error).

$$y^T \tilde{A} y \leq x^T \tilde{A} x. \tag{39}$$

*Proof of Lemma C.9*: First, note that

$$
\begin{aligned}
y^T A y - (b^*)^T A b^* &= \sum_{i \neq j} A_{i,j} \left( y_i \cdot y_j - b_i^* \cdot b_j^* \right) \\
&= \sum_{i \neq j} \left( A_{i,j} - \tilde{A}_{i,j} \right) \cdot \left( y_i \cdot y_j - b_i^* \cdot b_j^* \right) + \sum_{i \neq j} \tilde{A}_{i,j} \left( y_i \cdot y_j - b_i^* \cdot b_j^* \right) \\
&\leq 2 \cdot \sum_{i \neq j} \left( A_{i,j} - \tilde{A}_{i,j} \right) + \sum_{i \neq j} \tilde{A}_{i,j} \left( y_i \cdot y_j - b_i^* \cdot b_j^* \right)
\end{aligned}
$$

We will bound $\sum_{i \neq j} \left( A_{i,j} - \tilde{A}_{i,j} \right)$ by $\eta$, and $\sum_{i \neq j} \tilde{A}_{i,j} \left( y_i \cdot y_j - b_i^* \cdot b_j^* \right)$ by $\mathcal{E}rr$, which proves Lemma C.9.

**Bounding $\sum_{i \neq j} \left( A_{i,j} - \tilde{A}_{i,j} \right)$.** Denote $V_>$ the collection of $(\Delta, \eta)$-narrow vertices. Since the graph is $(\Delta, \eta)$-wide, we have For each $i \in V_>$,

$$\sum_{i \in V_>} \sum_{j \in [n]} A_{i,j} = \sum_{i \in V_>} W_i \leq \eta \cdot W. \tag{40}$$

Next, consider a $(\Delta, \eta)$-wide vertex $i$. It holds that $\deg(i) \geq \Delta$, and the maximum edge weight in its $\Delta$-suffix is bounded by $\eta \cdot W_i / \Delta$. Since

$$\tilde{A}_{i,j} = \min \left\{ A_{i,j}, \eta \cdot W_i / \Delta \right\}, \forall j \in [n], \tag{41}$$

$A_{i,j} - \tilde{A}_{i,j} = 0$ for each $j$ in the $\Delta$-suffix. Therefore, the gap $\sum_{i \neq j} \left( A_{i,j} - \tilde{A}_{i,j} \right)$ is at most the total weight of edges in its $\Delta$-prefix, which is at most $\eta \cdot W_i$. It follows that

$$\sum_{i \in [n] \setminus V_>} \sum_{j \in [n] \setminus \{i\}} \left( A_{i,j} - \tilde{A}_{i,j} \right) \leq \sum_{i \in [n] \setminus V_>} \eta \cdot W_i \leq \eta \cdot W. \tag{42}$$

**Bounding $\sum_{i \neq j} \tilde{A}_{i,j} \left( y_i \cdot y_j - b_i^* \cdot b_j^* \right)$.** First, based on Lemma C.12:

$$y^T \tilde{A} y - (b^*)^T \tilde{A} b^* = y^T \tilde{A} y - x^T \tilde{A} x + x^T \tilde{A} x - (b^*)^T \tilde{A} b^* \leq x^T \tilde{A} x - (b^*)^T \tilde{A} b^*.$$

It remains to bound $x^T \tilde{A} x - (b^*)^T \tilde{A} b^*$. First, observe that $b^*$ is also solution for the optimization problem specified in Equation (33). Since $x$ is the optimal one,

$$x^T \tilde{A} z + \left\| \tilde{A} z - \tilde{A} x \right\|_1 \leq (b^*)^T \tilde{A} z + \left\| \tilde{A} z - \tilde{A} b^* \right\|_1$$

Further, since $x \in \{-1, 1\}^n$,

$$x^T \tilde{A} x - x^T \tilde{A} z = x^T (\tilde{A} x - \tilde{A} z) \leq \|x\|_\infty \cdot \left\| \tilde{A} x - \tilde{A} z \right\|_1 \leq \left\| \tilde{A} x - \tilde{A} z \right\|_1.$$

Therefore,

$$
\begin{aligned}
x^T \tilde{A} x - (b^*)^T \tilde{A} b^* &= x^T \tilde{A} x - x^T \tilde{A} z + x^T \tilde{A} z - (b^*)^T \tilde{A} b^* \\
&\leq \left\| \tilde{A} x - \tilde{A} z \right\|_1 + x^T \tilde{A} z - (b^*)^T \tilde{A} b^* \leq (b^*)^T \tilde{A} z + \left\| \tilde{A} z - \tilde{A} b^* \right\|_1 - (b^*)^T \tilde{A} b^*.
\end{aligned}
$$

Taking expectation of both sides and observing that $\mathbb{E}[z_i] = b_i^*$ gives

$$\mathbb{E}\left[ x^T \tilde{A} x - (b^*)^T \tilde{A} b^* \right] \leq \mathbb{E}\left[ \left\| \tilde{A} z - \tilde{A} b^* \right\|_1 \right].$$

Applying Lemma C.11 finishes the proof.

*Proof of Lemma C.11.* First, note that $\forall i \in [n]$, it holds that

$$\mathbb{E}\left[z_i\right] = b_i^*,$$

$$\mathbb{E}\left[(\tilde{A}z)_i\right] = \mathbb{E}\left[\sum_{j \in [n]} \tilde{A}_{i,j} z_j\right] = (\tilde{A}b^*)_i.$$

Since the $b_i(e)$'s are at least pairwise independent, so are the $z_j$. Therefore,

$$\mathbb{V}\mathrm{ar}\left[(\tilde{A}z)_i\right] = \sum_{j \in [n]} \tilde{A}_{i,j}^2 \cdot \mathbb{V}\mathrm{ar}\left[z_j\right].$$

By Jensen's inequality

$$\mathbb{E}\left[\sqrt{\left|(\tilde{A}z)_i - (\tilde{A}b^*)_i\right|^2}\right] \leq \sqrt{\mathbb{E}\left[\left|(\tilde{A}z)_i - (\tilde{A}b^*)_i\right|^2\right]} = \sqrt{\mathbb{E}\left[\left|(\tilde{A}z)_i - \mathbb{E}\left[(\tilde{A}z)_i\right]\right|^2\right]} = \sqrt{\sum_{j \in [n]} \tilde{A}_{i,j}^2 \cdot \mathbb{V}\mathrm{ar}\left[z_j\right]}.$$

Finally,

$$\mathbb{E}\left[\left\|\tilde{A}b^* - \tilde{A}z\right\|_1\right] = \sum_{i \in [n]} \mathbb{E}\left[\left|(\tilde{A}b^*)_i - (\tilde{A}z)_i\right|\right] \leq \sum_{i \in [n]} \sqrt{\sum_{j \in [n]} \tilde{A}_{i,j}^2 \cdot \mathbb{V}\mathrm{ar}\left[z_j\right]}.$$

*Proof of Lemma C.10.* First, the variance of the $z_i$ satisfies

$$\mathbb{V}\mathrm{ar}\left[z_i\right] \in O\left(\frac{1}{\deg(i) \cdot \varepsilon^2}\right).$$

Therefore,

$$\mathcal{E}rr = \sum_{i \in [n]} \sqrt{\sum_{j \in [n]} \tilde{A}_{i,j}^2 \cdot \mathbb{V}\mathrm{ar}\left[z_j\right]} \in O\left(\frac{1}{\varepsilon} \sum_{i \in [n]} \sqrt{\sum_{j \in [n]} \frac{\tilde{A}_{i,j}^2}{\deg(j)}}\right).$$

Denote $V_>$ the collection of $(\Delta, \eta)$-narrow vertices. For each $i \in V_>$, since $\tilde{A}_{i,j} = 0$,

$$\sqrt{\sum_{j \in [n]} \frac{\tilde{A}_{i,j}^2}{\deg(j)}} = 0. \tag{43}$$

On the other hand, for each $(\Delta, \eta)$-wide vertex $i$, it holds that $\deg(i) \geq \Delta$, and

$$\max_{j \in [n]} \tilde{A}_{i,j} \leq \eta \cdot W_i / \Delta. \tag{44}$$

Therefore,

$$\sqrt{\sum_{j \in [n]} \frac{\tilde{A}_{i,j}^2}{\deg(j)}} \leq \sqrt{\sum_{j \in [n]} \frac{\tilde{A}_{i,j}^2}{\Delta}} \leq \sqrt{\sum_{j \in [n]} \frac{\tilde{A}_{i,j} \cdot \eta \cdot W_i}{\Delta^2}} \leq \frac{\sqrt{\eta} \cdot W_i}{\Delta}.$$

And

$$\sum_{i \notin V_>} \sqrt{\sum_{j \in [n]} \frac{\tilde{A}_{i,j}^2}{\deg(j)}} \leq \sum_{i \notin V_>} \sum_{j \in [n]} \frac{\sqrt{\eta} \cdot W_i}{\Delta} \leq \frac{\sqrt{\eta} \cdot W}{\Delta}.$$

$\square$

$\square$

*Proof of Lemma C.12.* We prove by induction on $i$ that

$$(y_{1:i-1}, y_i, x_{i+1:n})^T \tilde{A} (y_{1:i-1}, y_i, x_{i+1:n}) \leq x^T \tilde{A} x. \tag{45}$$

When $i = 1$, since $\tilde{A}_{1,1} = 0$, it holds that

$$f(y_1) = (y_1, x_{2:n})^T \tilde{A} (y_1, x_{2:n}) = y_1 \cdot \sum_{i \neq 1} 2 \tilde{A}_{1,i} x_i + C,$$

where $C$ does not depend on $y_1$. Therefore, $f$ is linearly in $y_1$, and either $f(-1) \leq f(x_1)$ or $f(1) \leq f(x_1)$ holds. We can pick $y_1 \in \{-1, 1\}$ so that $f(y_1) \leq f(x_1)$.

Via similar argument, for each $i > 1$, either:

$$(y_{1:i-1}, -1, x_{i+1:n})^T \tilde{A} (y_{1:i-1}, -1, x_{i+1:n}) \leq (y_{1:i-1}, x_{i:n})^T \tilde{A} (y_{1:i-1}, x_{i:n}) \leq x^T \tilde{A} x,$$

or

$$(y_{1:i-1}, 1, x_{i+1:n})^T \tilde{A} (y_{1:i-1}, 1, x_{i+1:n}) \leq (y_{1:i-1}, x_{i:n})^T \tilde{A} (y_{1:i-1}, x_{i:n}) \leq x^T \tilde{A} x.$$

hold. This proves our claim. $\square$

## D. Probability Facts

*Fact* D.1. Let $X_1, \ldots, X_n$ be 4-wise independent random variables with mean 0. Then for each $t \in \mathbb{R}_{\geq 0}$,

$$\Pr\left[\left|\sum_{i \in [n]} X_i\right| \geq t\right] \leq \frac{\sum_{i \in [n]} \mathbb{E}[X_i^4] + 6 \cdot \sum_{i \neq j} \mathbb{E}[X_i^2]\,\mathbb{E}[X_j^2]}{t^4}. \tag{46}$$

*Proof.* By Markov's inequality,

$$\Pr\left[\left|\sum_{i \in [n]} X_i\right| \geq t\right] \leq \frac{\mathbb{E}\left[\left|\sum_{i \in [n]} X_i\right|^4\right]}{t^4}. \tag{47}$$

On the other hand, since the $X_i$ are 4-wise independent mean zero random variables,

$$\mathbb{E}\left[\left|\sum_{i \in [n]} X_i\right|^4\right] = \mathbb{E}\left[\sum_{i \in [n]} X_i^4 + \binom{4}{2} \cdot \sum_{i \neq j} X_i^2 X_j^2 + 2 \cdot \binom{4}{2} \sum_{i \neq j \neq k} X_i X_j X_k^2 + \binom{4}{1} \cdot \sum_{i \neq j} X_i X_j^3\right] \tag{48}$$

$$= \sum_{i \in [n]} \mathbb{E}[X_i^4] + \binom{4}{2} \cdot \sum_{i \neq j} \mathbb{E}[X_i^2]\,\mathbb{E}[X_j^2]. \tag{49}$$

$\square$

## E. Proofs from Section 4

*Proof of Theorem 4.1.* We first claim that $S \leftarrow S_0 \cup S_1 \cup S_2$ is a vertex cover. This claim is straightforward: $S_0 \cup S_1$ forms a vertex cover for heavy-heavy and heavy-light edges, while $S_2$ covers light-light edges. Together, they cover all edges.

Next, combining Lemma 4.3, Lemma 4.4, and Lemma 4.5 yields:

$$\mathbb{E}[|S|] \leq \mathbb{E}[|S_0| + |S_1| + |S_2|] \tag{50}$$
$$\leq \mathbb{E}[|S_0 \cap \mathcal{C}| + |S_0 \setminus \mathcal{C}| + |S_1| + |S_2|] \tag{51}$$
$$\leq \mathbb{E}\left[|S_0 \cap \mathcal{C}| + |S_0 \setminus \mathcal{C}| + |S_1| + |\mathcal{C}_{<\Delta} \setminus (S_0 \cup S_1)| \cdot \left(2 - 2\frac{\log\log\Delta}{\log\Delta}\right)\right] \tag{52}$$
$$\leq \mathbb{E}\left[|S_0 \cap \mathcal{C}| + |\mathcal{C}_{<\Delta} \setminus (S_0 \cup S_1)| \cdot \left(2 - 2\frac{\log\log\Delta}{\log\Delta}\right)\right] + \varepsilon^{10} \cdot |\mathcal{C}| + 2 \cdot \varepsilon^{200} \cdot |\mathcal{C}_{\geq\Delta}| \tag{53}$$
$$\leq |\mathcal{C}| \cdot \left(2 - 2\frac{\log\log\Delta}{\log\Delta}\right) + \varepsilon^{10} \cdot |\mathcal{C}| + 2 \cdot \varepsilon^{200} \cdot |\mathcal{C}_{\geq\Delta}| \tag{54}$$
$$\leq |\mathcal{C}| \cdot \left(2 + 3 \cdot \varepsilon^{10} - 2\frac{\log\log\Delta}{\log\Delta}\right) = |\mathcal{C}| \cdot \left(2 - \Omega\left(\frac{\log\log 1/\varepsilon}{\log 1/\varepsilon}\right)\right). \tag{55}$$

$\square$

Proving Lemmas 4.3 to 4.5 require the following additional lemma.

**Lemma E.1.** *Let $v$ be a fixed vertex satisfying* $\deg(v) \geq \Delta$. *Then*

$$\Pr\left[m_v \neq \mathbb{1}_{[v \in \mathcal{C}]}\right] \leq \exp\left(-2 \cdot \deg(v) \cdot \varepsilon^2\right). \tag{56}$$

*Proof of Lemma E.1.* Without loss of generality, assume that $v \in \mathcal{C}$. For each $u \in N(v)$, let $X_u \in \{0, 1\}$ be the prediction of whether $v \in \mathcal{C}$, by the edge $e = (u, v)$. Then $m_v \neq \mathbb{1}_{[v \in \mathcal{C}]}$ if $\sum_{u \in N(v)} X_u \leq \deg(v)/2$. Note that

$\mathbb{E}\left[\sum_{u \in N(v)} X_u\right] = \deg(v) \cdot (1/2 + \varepsilon)$. By Hoeffding's inequality, when $deg(v) \geq \Delta$

$$\Pr\left[\sum_{u \in N(v)} X_u \leq \deg(v)/2\right] \leq \exp\left(-2 \cdot \frac{\deg(v)^2 \cdot \varepsilon^2}{\deg(v)}\right) = \exp\left(-2 \cdot \deg(v) \cdot \varepsilon^2\right).$$

$\square$

We are ready to prove Lemmas 4.3 to 4.5.

*Proof of Lemma 4.3.* Vertices from $V \setminus \mathcal{C}$ are added to $S_0$ when one of the following events occurs:

(1) There exists a vertex $v \in \mathcal{C}_{\geq \Delta}$ such that $m_v = 0$. In this case, Algorithm 1, line 12 adds $N(v)$ to $S_0$, which may include vertices from $V \setminus \mathcal{C}$.

(2) A vertex $v \notin \mathcal{C}$ with $\deg(v) \geq \Delta$ and $m_v = 1$ can also be added to $S_0$ in Algorithm 1, line 10. This occurs only if either:

   (a) $v$ has a neighbor $u$ with $\deg(u) < \Delta$; or
   (b) all its neighbors $u$ satisfy $\deg(u) \geq \Delta$, but at least one neighbor $u$ has $m_u = 0$. In this case, $v$ is also added to $S_0$ when Algorithm 1 adds $N(u)$ to $S_0$. *This is already accounted for in Case (1).*

To bound the number of vertices added to $S_0$ from $V \setminus \mathcal{C}$, it suffices to separately bound the numbers added from cases (1) and $(2.a)$.

*Case (1):* In this case, based on Lemma E.1, the expected number is bounded by

$$\mathbb{E}\left[\sum_{v \in \mathcal{C}_{\geq \Delta}} \mathbb{1}_{[m_v = 0]} \cdot \deg(v)\right] = \sum_{v \in \mathcal{C}_{\geq \Delta}} \Pr[m_v = 0] \cdot \deg(v) \tag{57}$$

$$\leq \sum_{v \in \mathcal{C}_{\geq \Delta}} \exp\left(-2 \cdot \deg(v) \cdot \varepsilon^2\right) \cdot \deg(v) \tag{58}$$

$$\leq |\mathcal{C}_{\geq \Delta}| \cdot \exp\left(-2 \cdot \Delta \cdot \varepsilon^2\right) \cdot \Delta \tag{59}$$

$$\leq \varepsilon^{10} \cdot |\mathcal{C}_{\geq \Delta}|, \tag{60}$$

where Equation (59) follows since the function $y = x \cdot \exp\left(-2 \cdot x \cdot \varepsilon^2\right)$ decreases when $x \geq 1/(2 \cdot \varepsilon^2)$, and Equation (60) follows since $\Delta \geq 1/\varepsilon^2$, for moderately small $\varepsilon$, it holds that $\exp\left(-2 \cdot \Delta \cdot \varepsilon^2\right) \cdot \Delta \leq \varepsilon^{10}$.

*Case (2.a):* We partition the vertices $v$ added to $S_0$ in the this case, into a collection of $|\mathcal{C}|$ subsets, denoted as $\{P_u : u \in \mathcal{C}_{<\Delta}\}$, as follows. Since $v \notin \mathcal{C}$, it holds that $N(v) \subseteq \mathcal{C}$, otherwise the edges incident to $v$ are not covered. Therefore, $N(v) \cap \mathcal{C}_{<\Delta} \neq \varnothing$. We pick an arbitrary vertex $u \in N(v) \cap \mathcal{C}_{<\Delta}$ and place $v$ into $P_u$. Based on Lemma E.1, we see that

$$\Pr[v \in P_u] \leq \Pr[m_v = 1] \leq \exp\left(-2 \cdot \Delta \cdot \varepsilon^2\right).$$

It remains to bound $\sum_{u \in \mathcal{C}_{<\Delta}} |P_u|$. Since each $|P_u|$ can be viewed as a sum of at most $\deg(u) \leq \Delta$ indicator random variables, each taking one with probability at most $\exp\left(-2 \cdot \Delta \cdot \varepsilon^2\right)$, it concludes that

$$\sum_{u \in \mathcal{C}_{<\Delta}} \mathbb{E}[|P_u|] = \sum_{u \in \mathcal{C}_{<\Delta}} \Delta \cdot \exp\left(-2 \cdot \Delta \cdot \varepsilon^2\right) \leq \varepsilon^{10} \cdot |\mathcal{C}_{<\Delta}|, \tag{61}$$

where the last inequality holds if $\varepsilon$ is moderately small.

*Case (2.b):* Note that such a vertex $v$ is already accounted for in Case (1) since it has neighbor $u \in \mathcal{C}_{\geq \Delta}$ with $m_u = 0$.

$\square$

*Remark* E.2. Indeed 4-wise independence in the prediction model of Definition 1.1 is enough to establish this proof. For example, let $v \in \mathcal{C}_{\geq \Delta}$, and we would like to prove that

$$\deg(v) \cdot \Pr\left[m_v = 0\right] \in o(1). \tag{62}$$

Let $X_u \doteq \mathbb{1}_{[b_v((u,v))=0]} - \left(\frac{1}{2} - \varepsilon\right), u \in N(v)$ be 4-wise independent random variables, where $b_v((u,v))$ is the prediction of $v$ from the edge $(u,v)$. Then

$$\mathbb{E}\left[X_u\right] = 0,$$

$$\mathbb{E}[X_u^2] = \left(\frac{1}{2} + \varepsilon\right)\left(\frac{1}{2} - \varepsilon\right),$$

$$\mathbb{E}[X_u^4] = \left(\frac{1}{2} + \varepsilon\right) \cdot \left(\frac{1}{2} - \varepsilon\right)^4 + \left(\frac{1}{2} - \varepsilon\right) \cdot \left(\frac{1}{2} + \varepsilon\right)^4 = \left(\frac{1}{2} + \varepsilon\right) \cdot \left(\frac{1}{2} - \varepsilon\right) \cdot 2 \cdot \left(\frac{1}{4} + \varepsilon^2\right) \leq \frac{1}{8}.$$

Based on Fact D.1, there exists some universal constant $c > 0$, s.t.,

$$\Pr\left[m_v = 0\right] \cdot \deg(v) = \Pr\left[\sum_{u \in N(v)} X_u \geq \deg(v) \cdot \varepsilon\right] \cdot \deg(v)$$

$$\leq \frac{\deg(v)/8 + 6 \cdot \deg(v) \cdot (\deg(v) - 1)/16}{\deg(v)^4 \varepsilon^4} \cdot \deg(v) \leq \frac{c}{\deg(v)\varepsilon^4}.$$

If we pick a threshold $\Delta$ properly, e.g., $\Delta \in \Omega(1/\varepsilon^{14}$, then $\deg(v) \geq \Delta$ implies that

$$\frac{c}{\deg(v)\varepsilon^4} \in O(\varepsilon^{10}). \tag{63}$$

*Proof of Lemma 4.4.* We will show that

(1) $\mathcal{C}_{\geq \Delta} \cup S_0$ covers all heavy-heavy and heavy-light edges, and

(2) $\mathbb{E}[|\mathcal{C}_{\geq \Delta} \setminus S_0|] \leq \varepsilon^{200} \cdot |\mathcal{C}_{\geq \Delta}|$.

Let $\mathcal{C}_1$ be the optimal vertex cover for the collection of edges that are incident to at most one vertex with degree $\geq \Delta$ and not covered by $S_0$. It follows that $|\mathcal{C}_1| \leq |\mathcal{C}_{\geq \Delta} \setminus S_0|$

$$\mathbb{E}[|S_1|] \leq \mathbb{E}[2 \cdot |\mathcal{C}_1|] \leq \mathbb{E}[2 \cdot |\mathcal{C}_{\geq \Delta} \setminus S_0|] \leq 2 \cdot \varepsilon^{200} \cdot |\mathcal{C}_{\geq \Delta}|. \tag{64}$$

*Proving (1):* Each heavy-heavy edge is trivially covered by $\mathcal{C}_{\geq \Delta}$, as at least one of its endpoints belongs to $\mathcal{C}$ and, consequently, to $\mathcal{C}_{\geq \Delta}$. For each heavy-light edge $(u,v)$, if one of its endpoints belongs to $\mathcal{C}_{\geq \Delta}$, then the edge is covered. Otherwise, without loss of generality, assume $\deg(v) \geq \Delta$ and $v \notin \mathcal{C}_{\geq \Delta}$. We claim that $(v,u)$ must already be covered by $S_0$.

Note that $v$ does not satisfy Line 7 of Algorithm 1, as one of its neighbors, $u$, has degree $\deg(u) < \Delta$. Therefore, Algorithm 1 either adds $v$ to $S_0$, or it adds $N(v)$ to $S_0$. In both cases, $(v,u)$ is covered by $S_0$.

*Proving (2):* Consider a fixed $v \in \mathcal{C}_{\geq \Delta}$. If $v \notin S_0$, then either

* $m_v = 0$, which happens with probability (according to Lemma E.1)

$$\Pr\left[m_v = 0\right] \leq \exp\left(-2 \cdot \deg(v) \cdot \varepsilon^2\right) \leq \exp\left(-2 \cdot \Delta \cdot \varepsilon^2\right). \tag{65}$$

* Or all its neighbors $u$ have degree $\geq \Delta$ and $m_u = 1$. In this case, at least one neighbor $u \notin \mathcal{C}$, as otherwise, we could remove $v$ from $\mathcal{C}$ while maintaining it as a valid cover. Hence,

$$\Pr\left[v \notin S_0\right] \leq \Pr\left[m_u = 1\right] \leq \exp\left(-2 \cdot \deg(u) \cdot \varepsilon^2\right) \leq \exp\left(-2 \cdot \Delta \cdot \varepsilon^2\right).$$

We conclude that

$$\mathbb{E}[|\mathcal{C} \setminus S_0|] = \sum_{v \in \mathcal{C}_{\geq \Delta}} \mathbb{E}[\mathbb{1}_{[v \notin S_0]}] \leq |\mathcal{C}_{\geq \Delta}| \cdot \exp\left(-2 \cdot \Delta \cdot \varepsilon^2\right) = |\mathcal{C}_{\geq \Delta}| \cdot \exp\left(-200 \cdot \ln(1/\varepsilon)\right) \leq |\mathcal{C}_{\geq \Delta}| \cdot \varepsilon^{200}.$$

$\square$

*Proof of Lemma 4.5.* By definition, $S_0 \cup S_1$ covers all heavy-heavy and heavy-light edges. Since $\mathcal{C}_{<\Delta}$ is a cover for the light-light edges, it also covers any edges not covered by $S_0 \cup S_1$. Furthermore, since $S_2$ is a $\left(2 - 2\frac{\log \log \Delta}{\log \Delta}\right)$-approximate cover for the edges not covered by $S_0 \cup S_1$, the claim follows. $\square$

