# OpenReview forum: "Improved Approximations for Hard Graph Problems using Predictions"
_ICML.cc/2025/Conference — ICML 2025 poster_

### Official Review · Reviewer_GWRK · 2025-03-06

**Overall Recommendation:** 2

**Summary:**

This paper addresses NP-hard graph problems by developing learning-enhanced approximation algorithms. The authors identify that existing prediction-based approaches predominantly rely on vertex-level information, which may limit performance improvements. To overcome this limitation, they propose a novel framework incorporating edge prediction mechanisms to enhance approximation algorithms. The core technical contribution involves systematically integrating edge prediction information with classical approximation methods through adaptive thresholding strategies. Experimental validation is conducted on two moderately-sized graphs: Facebook and Congress. Results demonstrate consistent performance improvements across varying ε parameters, compared to baselines.

**Claims And Evidence:**

Basically yes. However, it would be even better if the authors could address the following concerns of mine.

**Essential References Not Discussed:**

N/A.

**Experimental Designs Or Analyses:**

1. This paper only uses two graphs, but it does not explicitly address whether the algorithm remains efficient in terms of both time and space complexity when applied to very large graphs. This point is not clearly discussed.

2. This paper conducts experiments on two moderately-sized graphs, and although the results show that "for both datasets, we demonstrate that for an appropriate ε, our learning-augmented algorithm achieves the best performance on both graphs," it seems that the paper does not provide a clear explanation of how to determine this ε for graphs of different sizes.

**Methods And Evaluation Criteria:**

The benchmark datasets used in the paper do not reflect the essence of the problem. Firstly, according to the authors' description, the two datasets consist of medium-sized graphs, so the proposed algorithm has not been validated on graphs of other sizes. Secondly, the datasets used in the paper have a long time interval between them, and it seems that there are other datasets available in this field. Please provide a reasonable explanation.

**Other Comments Or Suggestions:**

N/A.

**Other Strengths And Weaknesses:**

## Strengths:
1. This paper introduces a learning-enhanced framework based on edge prediction, which offers a new perspective for improving approximation algorithms for NP-hard graph problems by incorporating predictive information.
2. This paper provides a detailed theoretical analysis, including a series of lemmas and proofs, such as Lemma A.5 to A.9, which lay a solid theoretical foundation for the algorithm's effectiveness and performance guarantees.

## Weaknesses:
1. The algorithm relies on a learning-enhanced framework, suggesting its performance depends on the accuracy of the predictions. The paper doesn't clearly explain how prediction errors are handled and whether they might significantly degrade the algorithm’s performance.

**Questions For Authors:**

Please answer the two questions in the "Experimental Designs Or Analyses" section as well as the question in "Weaknesses".

**Relation To Broader Scientific Literature:**

In the abstract, the authors state that their algorithm builds upon and extends the ε-prediction framework introduced by Cohen‐Addad, d’Orsi, Gupta, Lee, and Panigrahi (NeurIPS 2024). In the introduction, they also mention that 'learning-augmented algorithms have recently emerged as a popular paradigm for beyond worst-case analysis via incorporating machine learning into classical algorithm design,' thereby referencing related literature in this research area.

**Theoretical Claims:**

Yes.

---

> ### Author Rebuttal · Authors · 2025-04-01
>
> We thank the reviewer for their careful reading and comments. We address the weaknesses they mentioned below.
>
> **On datasets and Benchmarks**:
> The main focus of our paper is on giving rigorous theoretical improvements for classic NP-hard problems. We view our experiments as proof-of-concept, demonstrating that our theoretical ideas are also implementable. (We note that prior work such Cohen-Addad et al. who also studied augmenting NP-hard problems don't have experiments).
>
> We also provide some intuition for our experimental choices. For independent-set, we believed it is natural to test on social networks since the problem is optimizing for a collection of nodes with no mutual connections. Thus in our paper, we selected two social networks from the popular SNAP library. We picked moderately sized networks since we wanted to validate the quality of the approximation returned by our augmented algorithm by comparing to the exact optimal solution. However, the independent set is NP hard, so computing the exact optimal requires running an expensive integer linear program, which is prohibitive for large graphs (this is exactly why approximation algorithms for NP hard problems are useful!). Note that to use our algorithm, computing the optimum is not necessary; we only do it to compute our algorithm’s exact approximation factor on real world datasets. In practice, this step can be skipped since we already give a mathematical guarantee bounding the approximation factor.
>
> Lastly, **we ran a new experiment on a much larger graph** on a subgraph of large social network from SNAP (https://snap.stanford.edu/data/twitch_gamers.html) with ~50k nodes and ~1.1 million edges (we pruned the original graph to make the integer linear program for finding the optimal feasible). As seen in the figure in this anonymous link (https://ibb.co/60N2W8T1), the qualitative behavior remains the same: our learning-based algorithm can outperform the standard greedy approximation algorithm as well as the algorithm that only uses the predictions.
>
> >  does not explicitly address whether the algorithm remains efficient in terms of both time and space complexity when applied to very large graphs.
>
> All of our algorithms provably run in polynomial time and space so theoretically they are efficient for even very large graphs. This is our main message: by using very noisy predictions, we can get improved approximation algorithms for fundamental optimization problems in polynomial time. Our approximations using predictions overcome existing barriers which without predictions are not possible in polynoial time (assuming P != NP).
>
> >  the paper does not provide a clear explanation of how to determine this ε for graphs of different sizes
>
> > The paper doesn't clearly explain how prediction errors are handled
>
> Our guarantees **already have worst case guarantees built in**, even if the predictions are arbitrarily corrupt, in three ways.
>
> 1) Our approximation factors consist of two terms: one coming from the classic bounds without predictions and a term $f(\epsilon)$ that is the advantage that we have using edge-predictions that are correct with probability $½ + \epsilon$. (Note $f(\epsilon)$ depends on the problem). We recover the original worst-case guarantees by letting $\epsilon \rightarrow 0$. This corresponds to predictions that are random noise with no signal. On the other hand, our approximation factors improve as $\epsilon$ increases. Thus, our bounds naturally interpolate between the purely noisy case and to the case of large $\epsilon$ where we provably obtain an advantage over no predictions.
>
> 2) Even if the $\epsilon$ parameter is not known in practice, we can simply guess over multiple choices of $\epsilon$, run our algorithm, and take the best solution. E.g. in vertex cover, we can instantiate our algorithm for different $\epsilon$ values and take the smallest cover over all choices. This is because the problems we study are in NP, so we can compute the quality of the solution in polynomial time. Since for our theoretical bounds we only need to know $\epsilon$ up to a constant factor, our guess over $\epsilon$ can be done efficiently. That this also handles the reviewer’s other concern about determining $\epsilon$ in practice.
>
> 3) Lastly, we can also run another approximation algorithm in parallel to our algorithm, e.g. classic approximation algorithms, and take the best solution (this is because the problems are in the class NP so we can compute the quality of the solution returned by both algorithms). This ensures that we can never output something worse than the classic algorithm.
>
> Overall, we thank the reviewer for their feedback. We believe we have addressed the reviewer’s main concerns about how our algorithms scale, as well as how our algorithms handle prediction errors. We are happy to provide additional clarifications and engage further in the discussions if we have misunderstood any crucial points.
>
> Many thanks, The Authors.

---

### Official Review · Reviewer_CEUY · 2025-03-11

**Overall Recommendation:** 4

**Summary:**

This paper introduces a new prediction model that extends the framework of Cohen-Addad et al. (NeurIPS 2024) to improve approximation ratios for NP-hard graph problems, including (weighted and unweighted) Vertex Cover, Set Cover, Max Independent Set, and Max Cut. In their prediction model, each edge is assigned i.i.d. bits that provide $\epsilon$-accurate information about the variables participating in the edge constraint. For instance, in the Vertex Cover and Max Independent Set problems, each edge has two bits indicating whether its endpoints belong to a fixed optimal solution. Each bit is independently correct with probability $1/2+\epsilon$, regardless of the other bit or other edges. Using this prediction model, they achieve improved approximation ratios over classical approximation algorithms (without predictions) and learning-augmented approaches based on alternative prediction frameworks. The main algorithmic insight is to leverage predictions for high-degree vertices—where majority voting yields more reliable estimates—and apply a standard approximation algorithm for low-degree vertices.

**Claims And Evidence:**

All the claims are mathematically proved.

**Essential References Not Discussed:**

I am not aware of any related works that are essential to understanding the key contributions of the paper but are not currently cited.

**Experimental Designs Or Analyses:**

The experimental setup, baselines, datasets, and performance measure are reasonable. However, it would be better to also report the variance of the results for algorithms that use predictions.

**Methods And Evaluation Criteria:**

The paper evaluates the performance of its algorithms using the approximation ratio, which is standard in the literature. Additionally, its learning-augmented framework, though novel, is reasonable.

**Other Comments Or Suggestions:**

It would be interesting to see how the algorithms perform if the predictions are not $\epsilon$-accurate but instead come from a machine learning algorithm that does not have access to the current instance.

**Other Strengths And Weaknesses:**

The new prediction model introduced in this paper is interesting, achieves strong results, and improves upon previous results in other models. The proposed algorithms are simple and intuitive, and their ideas may have applications in other related problems. The paper is well-written and easy to follow. In particular, presenting the big picture and the main algorithmic ideas in the introduction was especially helpful.

**Questions For Authors:**

N/A

**Relation To Broader Scientific Literature:**

In Section 3, the authors compare their work with previous classical and learning-augmented algorithms for the studied problems. They also mention hardness results related to these problems. The previous learning-augmented results use different prediction frameworks; I mention two of them here:
 * Antoniadis et al. (2024) studied the Weighted Vertex Cover under a different prediction model that predicts the optimal set of vertices, and achieved an approximation ratio of $1+\frac{\eta^+ + \eta^-}{OPT}$, where $\eta^+$ and $\eta^-$ are the total weight of the false positive and false negative edges, respectively. In contrast, this work presents an algorithm with approximation factor $2-\Omega(\frac{\log \log 1/\epsilon}{\log 1/\epsilon})$.
 * Cohen-Addad et al. (NeurIPS 2024) studied the Max Cut problem under $\epsilon$-accurate vertex predictions, and achieved an approximation ratio of $\alpha_{GW}+\tilde{\Omega}(\epsilon^4)$, where $\alpha_{GW}$ is the Goemans-Williamson constant. In contrast, this work uses $\epsilon$-accurate edge predictions and achieves an approximation ratio of $\alpha_{GW}+\tilde{\Omega}(\epsilon^2)$.

**Theoretical Claims:**

I did not check the proofs in the appendices, but the proof sketches in the main body of the paper make sense.

---

> ### Author Rebuttal · Authors · 2025-04-01
>
> We thank the reviewer for their careful reading and comments.
>
> >  report the variance of the results for algorithms
>
> Thank you; we will do this in the final version.
>
>
> > It would be interesting to see how the algorithms perform if the predictions are not eps-accurate
>
> Thank you for this question. This is a nice future direction to study. However, we note that even if the $\epsilon$ parameter is not known in practice, we can simply guess over multiple choices of $\epsilon$, run our algorithm, and simply take the best solution. E.g. in the vertex cover example, we can instantiate our algorithm for different $\epsilon$ values and take the smallest vertex cover over all choices.

---

### Official Review · Reviewer_si3e · 2025-03-14

**Overall Recommendation:** 4

**Summary:**

The paper studies learning augmented algorithms for hard graph problems. The author introduce a new setting in which the algorithm can count on a prediction algorithm that provides two bits per edge, one per each incident vertex, which are positively correlated to the fact that the vertex satisfies the edge constraint. They show that in this setting algorithms can be designed whose approximation guarantees breaks the hardness barrier (in the absence of the predictions). Moreover they show that this model might be more powerful than the one where one bit per vertex is predicted.

## Update after rebuttal
I kept my positive score. The rebuttal phase did not bring any additional information to justify a decrease.

**Claims And Evidence:**

The claims are supported by proof of the stated improved bounds.
Moreover the authors present a moderate experimental analysis for the maximum independent set problem.
Here, I would have expected also a comparison with the algorithm of Braveman et al.

**Essential References Not Discussed:**

I think the treatment of the related literature is comprehensive

**Experimental Designs Or Analyses:**

See the above comment in Claims and Evidence

**Methods And Evaluation Criteria:**

Yes. The proofs appear to be sound.

**Other Comments Or Suggestions:**

a few typos and corrections:

page 3, lines 1 and 9, column 2:
 above which --> on which

page 3, line 11, column 2
 many --> may

page 4, line 10 column 1
 Minimum --> Maximum

Proof sketch of Theorem 4.1 (equation in display)
in the left hand side of the inequality: w(S) should be S
in the right hand side of the inequality, the first term should talk about S_0 not S_1

**Other Strengths And Weaknesses:**

Strenghts: a new model is proposed and its efficacy and imprevements are proved w.r.t. previous model.
A general approach for the new setting is designed that exploits known algorithms for degree-constrained instances.
Weaknesses: it is not clear how in practical cases one can make available predictions with the desired  bounded reliability.

**Questions For Authors:**

How do you guarantee the goodness of the prediction?
Can you provide experimental comparison with the other prediction-augmented algorithms proposed in the literature, rather than with artificial new prediction algorithms or non-prediction-based algorithm?

**Relation To Broader Scientific Literature:**

The paper introduces a new setting for prediction-augmented algorithms. While the previously proposed setting was based on one predicted bit per vertex, here more bits, namely one per incident edge, are predicted per each vertex.
The authors discuss both the significance and the positive gap in efficecncy between the new setting the previous one.

**Theoretical Claims:**

I checked all the proofs in the body of the paper. I did not verify the appendix.

---

> ### Author Rebuttal · Authors · 2025-04-01
>
> We thank the reviewer for their careful reading and comments.
>
> > a few typos and corrections:
>
> Thank you, we have fixed the typos.
>
> > It is not clear how in practical cases one can make available predictions with the desired bounded reliability.
>
> > How do you guarantee the goodness of the prediction?
>
> Our guarantees **already** have worst case guarantees built in, even if the predictions are arbitrarily corrupt, in three ways.
>
> 1) Our approximation factors consist of two terms: one coming from the classic bounds without predictions and a term $f(\epsilon)$ that is the advantage that we have using edge-predictions that are correct with probability $½ + \epsilon$. (Note $f(\epsilon)$ depends on the particular problem studied). We recover the original worst-case guarantees by letting $\epsilon \rightarrow 0$. This corresponds to predictions that are random noise and have no signal. On the other hand, our approximation factors improve as $\epsilon$ increases. Thus, our bounds naturally interpolate between the purely noisy case (where our guarantees converge to worst-case bounds) and to the case of large $\epsilon$ where we provably obtain an advantage over the setting with no predictions.
>
> 2) Even if the $\epsilon$ parameter is not known in practice, we can simply guess over multiple choices of $\epsilon$, run our algorithm, and simply take the best solution. E.g. in the vertex cover example, we can instantiate our algorithm for different $\epsilon$ values and take the smallest vertex cover over all choices. This is because the problems we study are in the class NP, so we can compute the quality of the solution in polynomial time. Since for our theoretical bounds we only need to know $\epsilon$ up to a constant factor, our guess over $\epsilon$ can be done efficiently. Note that this also handles the reviewer’s other concern about determining $\epsilon$ in practice.
>
> 3) Lastly, we can simply run another approximation algorithm in parallel to our algorithm, e.g. the classic approximation algorithms without predictions, and take the best solution (again this is because the problems are in the class NP so we can compute the quality of the solution returned by both algorithms). This ensures that we can never output something worse than the classic algorithms.
>
> > Can you provide experimental comparison with the other prediction-augmented algorithms proposed in the literature.
>
> For independent-set (the setting of our experiments) we are only aware of one prior work of Braverman et al. They use a different prediction model (vertex-based predictions) and obtain a substantially worse theoretical approximation factor (their approximation factor can be $\sqrt{n}$ whereas we get a constant approximation). The authors did not provide an implementation of their algorithm.
>
> Nevertheless, we tested their algorithm on the two datasets in our main submission. If we implement their algorithm as it is written in their paper, then their algorithm just converges to the standard greedy algorithm (dotted green line in our Figures). This is because their algorithm first prunes nodes based on a complicated degree condition (see Algorithm 1 in https://arxiv.org/pdf/2407.11364) and then runs the greedy solution on the pruned graph. However, the constants are quite large in the pruning condition so in the two graphs we tested, none of the nodes were pruned (and we suspect this to be the case for most “real world” graphs). Thus, their algorithm performs their step 4 (compute the greedy solution on the remaining un-pruned nodes) on the entire graph. (More precisely, their condition 2 of including all nodes in L with degrees at most $36 \cdot \log n$ always includes all nodes in our datasets). Perhaps one can optimize their constants to obtain a more reasonable bound, but we did not pursue this. Given this, we believe their algorithm to be mostly of theoretical interest, but it is an interesting direction for future work to devise a more practical version of their algorithm.

---

### Official Review · Reviewer_Q5LC · 2025-03-17

**Overall Recommendation:** 1

**Summary:**

They design algorithms for some fundamental NP-hard graph problems such as (Weighted) Vertex Cover, Set Cover, Maximum Independent Set, and MaxCut when we have some random information about an optimal solution. More precisely, they assume that for each edge, we have two bits for its endpoints regarding whether they are in the optimal solution or not, but this bit does not have the accurate data. Instead, each bit correctly reflects the optimal solution with probability $1/2 + \epsilon$. Their algorithms achieve a better approximation factor than those designed for the standard setting without predictions.

Their general idea is that for high degree vertices, decide their status in the solution based on the majority of information we have about them. This makes sense since if the degree of a vertex is $d$, then we have a $d$ sample about whether this vertex is inside the optimal solution or not. As the number of samples increases, the confidence in knowing the vertex's state improves. Finally, they handle low-degree vertices using a simple greedy approach specific to each problem.

**Claims And Evidence:**

They provide proof for their claims.

**Essential References Not Discussed:**

No.

**Experimental Designs Or Analyses:**

No.

**Methods And Evaluation Criteria:**

They provide an evaluation, but only for one of their proposed algorithms. However, this is not a major concern, as the main contribution of the paper lies in its theoretical results.

**Other Comments Or Suggestions:**

You mention previous work on vertex-based predictions but do not provide their approximation factors. Including these would facilitate comparison with your results.

**Other Strengths And Weaknesses:**

I am not sure why their model is interesting. Since the predictions are related to vertices, it does not make sense that each vertex has a separate prediction for each adjacent edge. The prior model, where each vertex had a single prediction bit, seems more realistic to me. Additionally, this model provides excessive information about high-degree vertices, which their algorithm takes advantage of. I find the paper's motivation insufficient in justifying the significance of their model.

**Questions For Authors:**

N/A

**Relation To Broader Scientific Literature:**

Previous works study the case that for each vertex we have one random bit that gives information about the optimal solution. In contrast, this work assigns two bits per edge, providing significantly more information for high-degree vertices.

**Theoretical Claims:**

I reviewed the proofs in the main part of the paper, but not the appendix.

---

> ### Author Rebuttal · Authors · 2025-04-01
>
> We thank the reviewer for their careful reading and comments. We address the weaknesses they mentioned below.
>
> > Their general idea is that for high degree vertices, decide their status in the solution based on the majority of information we have about them ... Finally, they handle low-degree vertices using a simple greedy approach specific to each problem.
>
> We agree that our high level idea of separating the vertices into heavy and light degrees is quite intuitive and generalizes across many problems. We view this as a strength of our framework. However, we would like to point out the many technical challenges we need to overcome.
>
> Our ‘high degree’ threshold is a constant (depending on $\epsilon$ and independent of the graph size). This alone is not enough to union bound over potentially $O(n)$ high-degree vertices. Thus, there is always a non-negligible chance that a high-degree vertex gets misclassified. This can be very problematic since high-degree vertices can be highly influential in the final solution. For example in vertex cover, a high-degree vertex which is misclassified to not be in the solution can make us add all of its neighbors in the vertex cover, leading to an unbounded competitive ratio if we are not careful. Thus in all of our problems, we need a sophisticated “cleaning step” which not only fixes any misclassifications, but also helps us successfully merge the solution found for low-degree vertices. This is non-trivial since the solution on low-degree vertices may conflict with the solution on high-degree vertices, e.g. for independent set. Our cleaning step is especially subtle for the weighted vertex cover problem,where  the weight of a vertex is unrelated to its degree (see the paragraph starting on line 589 for a technical discussion).
>
> > I am not sure why their model is interesting.
>
> We believe the edge-based prediction model that we introduce is interesting for the following reasons:
>
> 1) Our model gives much stronger theoretical results compared to vertex predictions. For example in Max Cut, vertex predictions in Cohen-Addad et al. give $\approx \epsilon^4$ additive approximation advantage over the best classical approximation whereas edge predictions give $\approx \epsilon^2 \gg \epsilon^4$ advantage. The difference is much more pronounced for independent set where we can get a constant factor approximation, whereas prior work Braverman et al. can only guarantee a $O(\sqrt{n})$ factor approximation with vertex predictions. For vertex cover (weighted and unweighted), we can get strictly smaller than a factor of 2 approximation whereas the vertex prediction based algorithm of Antoniadis et al. has an approximation factor depending on the number of predicted false positives and negatives, which can lead to an unbounded approximation ratio.
>
> 2) Our results do not require i.i.d. predictions across edges. Rather, 4-wise independence of the predictions suffices (see our Remark 4.6). This means that our algorithms can handle potentially a huge number of correlations among the predictions.
>
> 3) Lastly, while our work is the first to introduce edge predictions for augmenting NP-hard problems, we remark that edge based predictions have also been used in other learning-augmented optimization problems (unrelated to NP-complete problems), e.g. [1] for correlation clustering and [2] for metric clustering.
> - KwikBucks: Correlation Clustering with Cheap-Weak and Expensive-Strong Signals. ICLR ‘23
> - Metric Clustering and MST with Strong and Weak Distance Oracles. COLT ‘24.
>
> > You mention previous work on vertex-based predictions but do not provide their approximation factors. Including these would facilitate comparison with your results.
>
> Please see the first point of our response above. We also note that our submission does contain a thorough discussion on prior work on vertex-based predictions. See Lines 146-164 (right column) for discussion on prior work on vertex-cover, Lines 174-185 (left column) for independent set, and Lines 186-190 for discussion of prior work on max-cut.
>
> Overall, we thank the reviewer again for their feedback. We believe we have addressed the reviewer’s main concern about why our new model is interesting. We are happy to provide additional clarifications and engage further in the discussions if we have missed or misunderstood any crucial points.
>
> Many thanks,
> The Authors.

---

### Decision · Program_Chairs · 2025-05-01

**Decision:**

Accept (poster)

**Comment:**

This submission studies learning-augmented approximation algorithms for a variety of classic NP-hard graph problems, including (Weighted) Vertex Cover, Set Cover, Maximum Independent Set, and MaxCut. The key innovation is to use edge-based predictions, where each edge provides bits of (potentially noisy) information about its endpoints’ membership in an optimal solution. The authors demonstrate that this edge-based model enables better approximation factors than are possible under traditional worst-case analyses or prior vertex-based prediction models. They also provide theoretical bounds indicating that even if predictions are arbitrarily inaccurate, their algorithmic approach recovers the standard worst-case approximation guarantees.

Despite one reviewer expressing skepticism about the new prediction model’s practicality, the majority of reviewers find the paper’s theoretical contributions compelling, with two clear accepts and one borderline evaluation. The rebuttal provides a thorough explanation of the edge-based prediction model’s advantages and clarifies strategies for handling prediction errors, underscoring the novelty and strength of the results.

From my (Area Chair) perspective, the paper’s contributions to learning-augmented approximation are significant and of interest. I do not share the concerns with the reviewer regarding the prediction model and, in fact, find the model natural and worth studying.